# Can the Introduction of Different Olive Cakes Affect the Carcass, Meat and Fat Quality of Bísaro Pork?

**DOI:** 10.3390/foods11111650

**Published:** 2022-06-03

**Authors:** Ana Leite, Rubén Domínguez, Lia Vasconcelos, Iasmin Ferreira, Etelvina Pereira, Victor Pinheiro, Divanildo Outor-Monteiro, Sandra Rodrigues, José Manuel Lorenzo, Eva María Santos, Silvina Cecilia Andrés, Paulo C. B. Campagnol, Alfredo Teixeira

**Affiliations:** 1Mountain Reserach Center (CIMO), Polytechnic Instituto f Bragança, Campus de Santa Apolónia, 5300-253 Bragança, Portugal; anaisabel.leite@ipb.pt (A.L.); lia.vasconcelos@ipb.pt (L.V.); iasmin@ipb.pt (I.F.); etelvina@ipb.pt (E.P.); srodrigues@ipb.pt (S.R.); 2Food Technology, Faculty of Sciences Ourense, University of Vigo, 32004 Ourense, Spain; jmlorenzo@ceteca.net; 3Research, Meat Technology Centre of Galicia (CTC), Rua Galicia No. 4, Parque Tecnológico de Galicia, 32900 San Cibrao das Viñas, Spain; rubendominguez@ceteca.net; 4Veterinary and Animal Reserach Centre (CECAV), University of Trás-os-Montes and Alto Douro, 5000-801 Vila Real, Portugal; vpinheir@utad.pt (V.P.); divanildo@utad.pt (D.O.-M.); 5Chemistry Academic Area, Autonomus University of the State of Hidalgo, Carr. Pachuca-Tulancingo Km 4.5 s/n, Col. Carboneras, Mineral de la Reforma, Pachuca 42183, Mexico; emsantos@uaeh.edu.mx; 6Center for Research and Development in Food Cryotechnology (CIDCA, CONICET-CICPBA-UNLP), National University of La Plata UNLP, 47 y 116, La Plata 1900, Argentina; scandres@biol.unlp.edu.ar; 7Department of Food Science and Technology, University of Santa Maria—USM, Santa Maria 97105-900, Brazil; paulocampagnol@gmail.com

**Keywords:** circular economy, olive industry by-product, animal feed, Bísaro pig diet, LTL muscle quality

## Abstract

The present study aimed to evaluate the effect of the inclusion of different olive cakes in the diet of Bísaro pigs on the carcass, meat and fat. The carcasses of 40 animals fed a diet with five treatments (T1—Basic diet and commercial feed; T2—Basic diet + 10% crude olive cake; T3—Basic diet + 10% olive cake, two phases; T4—Basic diet + 10% exhausted olive cake; T5—Basic diet + 10% exhausted olive cake + 1% olive oil) were used to study the effect on carcass traits, physicochemical meat quality and lipid composition of meat and backfat. There were no significant differences between treatments for the conformation measurements performed, except for the length at the seventh and last rib (*p* < 0.05). The percentage of prime cuts of the carcass in Bísaro pig is within the values indicated by the Portuguese Standard 2931. No significant differences between treatments for body weight, pH and carcass weight were found. The values of ultimate pH (5.7), *L** (51–52), *b** (11–12) and SF (3.4–4.2) observed confirm a non-exudative and firm meat without quality deviations, such as DFD or PSE. Thus, as a general conclusion, the inclusion of different olive cakes in the diet of Bísaro pigs did not cause any negative consequences on the carcass characteristics and conformation as well as in the meat and lipidic quality. In addition, the inclusion of this olive industry by-product in the animal diet would be an important contribution to solving the problem of the great environmental impact from olive-mill wastewaters from the extractive industries.

## 1. Introduction

The Portuguese olive sector, in particular the oil extractive industry, produces large amounts of olive by-products, considered extremely toxic, with a high negative environmental impact and causing great social concerns in rural areas. Olive cake is one of the most relevant by-products of the olive extraction industries, mainly constituted by olive stones, pulp, skin and water from olives and water added in the extraction process and during the malaxation phase [1,2,3,4,5,6]. There are several types of olive cakes depending on whether the extraction system has two phases or three phases. The two-phase system is more ecofriendly, as the olive cake is more moist and contains lower oil content, due to the more efficient separation of the oil from the other elements by the centrifugation system. The three-phase olive cake is drier, as during the extraction of olive oil, the olive paste is separated into three parts: olive oil, dry olive cake and olive mill wastewaters. The olive cakes can also be characterized by their composition and oil content, as crude olive cake and extracted olive cake [1,2,3,4]. Several strategies were developed to valorize and use olive by-products, namely their use as a thermal material to produce heat and electricity (mainly leaves, dry olive cake and olive wood from pruning) [7], for food industry purposes [8], cosmetics [9] and as animal feed [10,11]. The use of this by-product in animal feed is related to the fatty acid composition of the olive cake, with a high proportion of oleic acid and polyunsaturated fatty acids [1,12] as the incorporation of the olive cake in the swine diet could improve the growth performance and decrease carcass fat thickness [13] or, in ruminants, provide more energy [14]. Recently, and in accordance with the various agreements of the European Commission and legislated by the Portuguese Government [15], studies have indicated that the inclusion of olive by-products in animal feed has the potential for greater valorization. Furthermore, in recent years the significant increase in prices of feed raw materials based on cereals has caused a significant increase in the production costs for producers [13,16]. In compliance with current legislation that strongly encourages the food industry to find new uses for by-products [17], pigs are an important species for which to valorize wastes and by-products from the olive oil extraction process, particularly the Bísaro pig, a local rare breed (Celtic type) raised mainly in the north of Portugal, and almost extinct in the 1980s. During the last 30 years, the Bísaro breed has attracted increasing interest in traditional pig farming systems and sustainable production of premium meat products with protected designation of meat products (PDO) and protected geographical indication (PGI) [18]. The use of by-products from the olive oil supply chains can be a way to reduce food costs and, above all, reduce the negative environmental impact of the olive oil extraction industry. Additionally, some authors observed that the increase in olive cake in the diet promoted a reduction in saturated fatty acid (SFA) and increased the monosaturated fatty acids (MUFAs), improving the fat composition in pigs [13,19]. In addition, the nutritional quality of fresh meat from these pigs can be improved by reducing undesirable (saturated) fatty acids and increasing unsaturated fatty acids.

Thus, the main objectives of this study were the valorization of a by-product of the oil industry in the feeding of Bísaro pigs, evaluating the potentiality of inclusion of different olive cakes (crude olive cake, exhausted olive cake without and with 1% olive oil and two-phase olive cake) and its effect on carcass traits and meat and lipid quality of meat and backfat.

## 2. Materials and Methods

### 2.1. Animals and Diets

Forty Bísaro pigs weighing around 100 kg body weight (21 females and 19 males) were randomly divided into five groups and fed for 90 days with four different olive oil cakes in combination with a basic diet. Analysis of diets was performed at the Meat Technology Center of Galicia, Ourense, Spain, and the data are shown on Table 1.

The animals were separated into batches of 8, with the same age and feeding all treatments started at the same time and in the same conditions (feed level was “ad libitum” with an average consumption of 3 kg per day). Olive cakes were incorporating in the diet assuming a 10% addition as the starting point according to [5]. Different types of olive oil cakes were used from different extraction units that receive olives from all over the northeast of Portugal. The experimental feed trial was carried out at Trás-os-Montes e Alto Douro University, Vila Real, Portugal.

### 2.2. Slaughter Procedure

A total of 40 pigs were slaughtered at an average body weight of 135 ± 4.5 kg in the Municipal Slaughterhouse of Bragança. The slaughter procedure and carcass fabrication were previously described by Álvarez-Rodríguez and Teixeira [20]. All animals were cared for and slaughtered in compliance with the welfare regulations and respecting EU Council Regulation (EC) No. 1099/2009 [21].

After slaughter, the carcasses were placed in a cooling chamber. After cooling at 4 °C for 24 h, the carcass weight (CW) was recorded. The dressing percentage was expressed as the weight of the carcass in relation to body weight (CW/BW × 100). The head was separated at the atlas joint and weighed.

### 2.3. Carcass Quality Measurements

Carcass conformation measurements were assessed with a metal measuring tape on the carcass suspended in a gamble of constant width between legs [22]. Backfat thickness measurements were taken, not including the skin.

The following measurements were recorded:

Carcass length (L): this distance is measured in a straight line from the cranial edge of the manubrium of the sternum to the cranial edge of the pubic bone (A–B Figure 1, midsagittal).

Leg length (LL): distance from the tarsal–metatarsal joint surface to cranial edge of the pubic bone (Figure 1 midsagittal, A–C distance)

Width of the buttocks (d): maximal length between both greater trochanters of the femur (d–d′ distance Figure 1, dorsal).

Width of the thorax (e): the greatest width of the chest of the carcass at the level of the caudal edge of the scapula (e–e′ distance, Figure 1, dorsal).

The following measurements of backfat thickness were also recorded:

Backfat P1: thickness of the dorsal fat 4.5 cm from the dorsal midline of the vertebral column at the level of the last rib.

Backfat P2: thickness of the dorsal fat 6.5 cm from the dorsal midline of the vertebral column at the level of the last rib.

Backfat P3: thickness of the dorsal fat 8 cm from the dorsal midline of the vertebral column at the level of the last rib.

The carcass and muscle depth, muscle length and fat thickness were measured using the methodology described by Teixeira et al. [23].

### 2.4. Physicochemical Analysis of Meat and Fat

The pH was measured after slaughter and 24 h after slaughter (ultimate pH) according to the Portuguese standard [24], using a portable potentiometer equipped with a specific electrode penetrator, and calibrated with standard buffers with the following pH: 4.01 and 7.02.

Meat color was recorded over the surface of the *Longissimus thoracis et lumborum* (LTL) muscle after at least 15 min of blooming time for the pigments at the surface to oxygenate using the lightness (*L**), red-greenness (*a**) and yellow-blueness (*b**) system with a colorimeter (Lovibond RT Series Model SP62, Tintometer Inc., Sarasota, FL, USA). This color system was described with the coordinates *L* a* b** [25]. The color attributes identified as tom (H*) and chroma (C*) were measured according to the following equations:$$H=\tan^{-1}\left(\frac{b^{*}}{a^{*}}\right)$$
$$C=\sqrt{{\left(a^{*}\right)}^{2}+{\left(b^{*}\right)}^{2}}.$$

Color and pH procedures were carried out according to [26].

Then, the carcasses were carefully halved, and the left side was weighed (LCSW). Carcasses were carried to the Laboratory of Carcass and Meat Quality at the Agrarian School of the Polytechnic Institute of Bragança for carcass evaluation and meat analysis.

The carcass joints corresponding to the ham, shoulder, loin and neck were separated and weighed separately. LTL muscle and backfat samples were taken for physicochemical and fatty acid analysis in triplicate. Portuguese standard procedures were used for the determination of moisture [27], ash [28] and protein [29] contents. Water-holding capacity (WHC) was evaluated according to the Honikel procedure [30]. Shear force (SF) was evaluated, using an INSTRON 5543J-3177 equipped with a Warner–Bratzler device. Muscle samples of LTL *muscle* (100–120 g) were cooked inside plastic bags in a 70 °C water bath until reaching 70 °C in the muscle center. Half an hour later, muscle subsamples (1 cm^2^ cross-section) were taken from each muscle for SF evaluation. For each muscle sample, eight shear subsamples were taken. The measurement was recorded as the average yield force in kilograms (Kgf), required to shear perpendicularly to the direction of the fibers. All procedures were carried out at room temperature according to [31]. The collagen content was estimated by measuring the hydroxyproline concentration following the Portuguese Standard NP 1987 [32]. Heme pigments were obtained using the reflectance of the exposed surface by spectroscopy using a Spectronic Unicam 20 Genesys (SPECTRONIC 20 GENESYS, Thermo Fisher Scientific, Austin, TX, USA). The method is based on the muscle pigment content method by Hornsey [33] and results are expressed in mg myoglobin/g fresh muscle.

Fatty acids of the diets were analyzed according to the Folch procedure [34] and separation and quantification of the FAME were carried out using a gas chromatograph, GC-Agilent 6890N (Agilent Technologies Spain, S.L., Madrid, Spain), equipped with a flame ionization detector and an HP 7683 automatic sample injector.

Fatty acids in both loin and backfat samples were analyzed in the Carcass and Meat Quality Laboratory of ESA-IPB. The total lipids were extracted from 25 g of the meat sample according to the Folch procedure [34]. Fifty micrograms of fat were used to determine the fatty acid profile. The fatty acids were transesterified according to the method described by other authors [35]; 4 mL of a sodium methoxide solution was added, and vortexed every 5 min for 15 min at room temperature, then 4 mL of H_2_SO_4_ solution (in methanol at 50%) was added and vortexed briefly. Then, 2 mL of distilled water was added and it was vortexed again. The organic phase (with the methyl esters of fatty acids) was extracted with 2.35 mL of hexane. The FAME separation and quantification were performed using a gas chromatograph (GC-Shimadzu 2010Plus; Shimadzu Corporation, Kyoto, Japan) equipped with a flame ionization detector and an AOC-20i automatic sample injector and using a Supelco SP TM -2560 fused silica capillary column (100 m length, 0.25 mm i.d., 0.2 µm film thickness). To assess the lipid quality, the index of atherogenicity (*IA*) and the index of thrombogenicity (*IT*) according to Ulbricht and Southgate [36] were used:$$IA=\frac{C12:0+4\;X\;C14:0+C16:0}{\Sigma \mathrm{MUFA}+\Sigma \mathrm{PUFA}}$$
$$IT=\frac{C14:0+C16:0+C18:0}{\left(0.5\times \Sigma \mathrm{MUFA}\right)+\left(0.5\times n-6\right)+\left(3\times n-3\right)+\frac{n-3\;}{n-6}}.$$

### 2.5. Statistical Analysis

Data were analyzed using the statistical package JMP^®^ Pro 16.0.0 by 2021 SAS Institute Inc.^©^ (Cary, NC, USA). The predicted means obtained were ranked based on pair-wise least significance differences and compared using the Tukey´s HSD test for the *p* < 0.05 significance level.

## 3. Results and Discussion

### 3.1. Carcass Evaluation

The carcass conformation measurements are shown in Table 2. No significant differences (*p* ≥ 0.05) were observed between leg measurements and backfat thickness for the different treatments studied. All conformation measurements in all treatments confirm a carcass with an average length of 90 to 92 cm, a depth of 41 to 43 cm and buttock width and chest width of around 36 and 35 cm, respectively. The carcass length measurement found by Freitas et al. [37] for the Alentejano breed was considerably smaller than that recorded for Bísaro (15 cm). As the animals had a similar carcass weight, around 110 kg, the Bísaro carcasses are longer than those of the Alentejano breed. Martins et al. [38], in a study comparing carcass and meat quality from Alentejano and Bísaro breeds, recorded a longer carcass length for Bísaro (105 cm) than for Alentejano (89 cm). The leg length of around 60.88–65.61 cm was shorter than that obtained for the Celta breed and its crosses with another breed [39]. The backfat thickness varied from 5.38–5.46 cm in the cervical region to 4.39–5.17 cm in the lumbar region. At the same slaughter weight, the backfat thickness recorded in Alentejano pork [37] at the 3rd–4th rib (5.3–5.6 cm) was similar to that found in our work. The leg fat thickness varied between 7.64 and 8.66 cm in T3 and T4 treatments, respectively. Differences between studies indicate that these pigs are from traditional breeds (not meat improved breeds) with great variability in carcass shape and size, at the same mature body weight.

Figure 2 and Figure 3 show the measurements performed on the seventh rib and 13th and 14th rib. No significant differences were observed in the depth at the level of the 7th rib and the mean values obtained varied between 3.59 and 4.09 cm. Muscle length in the seventh rib was significantly longer (*p* < 0.05) for the T5 treatment compared to the others. No significant differences (*p* ≥ 0.05) were observed for fat thickness, which ranged from 5.35 to 5.65 cm. Fat thickness values much lower than those recorded by us, ranging between 1.79 and 1.90 cm, were observed for the Pietrain breed and for its cross with Duroc [40].

Backfat thickness measurements assessed at three standard points of the 13th and 14th rib (P1, P2 and P3) did not show significant differences between treatments (*p* ≥ 0.05) (Figure 3). Mean values between 3.65 and 4.31 cm were lower than those reported by Parunovic et al. [41] for the Mangalist breed (5.14 cm and 5.80 cm). However, lower values of fat thickness were still observed in the last rib in crossbred barrows (Duroc × Landrace × Large) [42]. Muscle length at the level of the 13th and 14th ribs showed significant differences, with the highest value observed in the T4 treatment (9.92 cm).

Figure 4 shows the yield (%) of the carcass joints considered as superior quality for the industry to process as cured products (ham, shoulder, loin and neck). No significant (*p* > 0.05) differences were observed between treatments. According to Malgwi et al. [43], it is difficult to compare the yield of carcass joints between studies because there are different jointing and commercial cutting procedures in different countries. Even so, the ham yield around 24% was relatively higher than those found by García-Casco et al. [16] in Iberian pigs fed with dry olive pulp and wet crude olive cake (21.94 and 22.34%, respectively) and slaughtered at 20 kg of body weight more than in the present study. However, the shoulder yield varying between 14.34 and 15.21% was moderately lower than that observed by García-Casco et al. [16] in Iberian pigs (15.99–16.15%). The ham and shoulder yields were higher than those obtained by Alvarez-Rodriguez and Teixeira [21], also in Bísaro pigs, but fed without the addition of olive cake in the diet. The loin yield between 3.50 and 3.87% was basically the same as the 3.2% observed in Iberian pigs [16]. The neck yield (corresponding to the cranial portion of the loin), nowadays too much appreciated by the meat processing industry and consumers, varied between 2.39 and 2.78%. In any case, the percentage of prime cuts of the carcass in Bísaro pork is within the values indicated by the Portuguese Standard 2931 [44] and the values found for Alentejano pork at the same carcass weight [37]. As a general and important conclusion, in the present study, the addition of olive industry by-products in the pigs’ diet had no significant influence on the yield of Bísaro carcass joints.

The effect of the different olive cake treatments on carcass characteristics of Bísaro pigs is shown in Table 3. No significant differences (*p* ≥ 0.05) were observed for BW nor for CW or LCSW. The carcass pH values showed correct decrease kinetics between slaughter and after 24 h of refrigeration and no differences were found between treatments. The values of pH and meat quality kinetics during carcass chilling followed the trend observed for breeds close to the Bísaro such as Celta and Asturcelta [45,46,47,48] and Alentejano [38]. The ultimate pH around 5.7–5.8 was a value within the normal range of 5.4–5.8 suggested by Fisher [49]. The dressing percentage for all treatments was around 79% and similar to that reported before by Álvarez-Rodríguez and Teixeira [21], in Iberian pigs by Garcia-Casco et al. [16], in Pietrain pigs [5], crossbred barrows (Duroc × Landrace × Large) [42] and for the Celta breed by Franco et al. [44] as well as in Alentejano pigs [34] or Alentejano and Bísaro and their crosses [38]. Moreover, the dressing percentage observed is within the average of 74% indicated for pigs [50].

In addition, the CIELAB color coordinates and attribute parameters did not show significant differences between treatments. The luminosity index (*L**), red index (*a**), yellow index (*b**) as well as the chroma (C*) and tom (H*) color attributes associated with the quality pH kinetics confirm that the carcasses were normal without quality deviations of type dark, firm and dry (DFD) or pale, soft and exudative (PSE). Compared to other studies on Bísaro carcasses [46] and commercial crosses with Duroc and Yorkshire boars [51], the luminosity index observed in this work is similar, but the red index is much higher. The values of the red index were like those observed in the Celta pig by other authors [52]. These differences could be explained by the body weight between both studies which was higher in the present study and, consequently, at an older age, this would in a more mature meat with a great myoglobin content and probably also a greater amount of heme pigments (as can be seen in Table 4). However, in Pietrain pigs [5], higher *L** values (about 57) and lower *a** values (7) were found than those found by us in Bísaro pigs, but with a lower body weight at the slaughter of Pietrain pigs. Martins et al. [38] reported lower values of *L**, *a** and *b** in Bísaro than in Alentejano pigs. Additionally, the mentioned authors found a similar value of *L** for the Bísaro pigs (50) but the value of *a** and especially *b** were much lower. The meat color is one of the most important quality parameters that determine the preference of meat by consumers and a red color is associated with the freshness of meat [53]. In our study, the addition of olive cake in the diets did not significantly affect the meat color. Dal Bosco et al. [54], studying the effect of supplementation with olive pomaces in the diet of rabbits, also did not find any difference in color parameters.

The WHC and SF parameters are important in terms of juiciness and tenderness [47]. Significant differences were observed in the mean values between treatments (*p* < 0.05) for WHC. The T3 had a significantly higher WHC value than T4 (18.45 and 12.11, respectively) and the other treatments were not significantly different between them. The values obtained in this study were much lower than those obtained by other authors in Korean Native Black pig and crossbred with Duroc pig [55], in Argentina, commercial crosses of hybrid females with Duroc and Yorkshire boars [51], Celta pig [47] and Alentejano and Bísaro [38]. Regarding shear force, no significant differences were found between treatments and varied from 3.42 to 4.17 kgf. Similar SF values were found for crossbred barrows (Duroc × Landrace × Large) (3.5) [42], Alentejano (4.2), and Bísaro (5.1) [38]. Lower values between 2 and 3.6 kgf were reported in Celta pigs [39].

The information of CIELAB color, in association with other physical carcass parameters, particularly the kinetics of the decline of pH and, according to van der Wal et al. [56], the values of ultimate pH (5.7), *L** (51–52), *b** (11–12) and SF (3.4–4.2), observed for Bísaro confirms a non-exudative and firm meat without quality deviations, such as dark, firm and dry (DFD) or pale, soft and exudative (PSE).

### 3.2. Chemical Composition

The chemical composition of LTL muscle for the five different treatments are shown in Table 4. No significant differences (*p* ≥ 0.05) were found for ash, heme pigment, collagen or protein contents. The pigs fed with crude olive oil (T2) presented the highest intramuscular fat proportion (7.22%), significantly higher (*p* < 0.05) than the control (T1) but not the other treatments. The effect of olive cake added to the diet on the fat deposition was more evident in the pork carcasses than that observed in rabbits [54]. The fat deposition observed in Bísaro pigs was higher than that recorded in Preto Alentejano males (3.8%), and commercial Large White males (3.3%) [57]. Liotta et al. [5] in a trial with Pietrain pigs fed with different proportions of olive cakes (50 and 100 g/kg) recorded a lower intramuscular fat percentage than that observed in the present study. Compared to the results obtained in our study, the cited authors observed a reduction in intramuscular fat in the pigs fed with olive cakes in relation to the control. The average values recorded in this study for the intramuscular fat (4.86 and 7.22%) were within the range of those observed for the Celta breed (5.22%) but higher than the cross between Celta and Duroc (3.96%) and the cross of Celta and Landrace (3.08%) [39]. Values of intramuscular fat within the interval of the present study were reported for the Alentejano pig (7%) and for the Bísaro pig (6%) [38].

As expected, the olive cake added to the diet has an inverse effect on moisture compared to intramuscular fat content. The T2 treatment presented a significantly (*p* < 0.05) lower percentage (67.55%) than the control (70.10%) but no significant differences were found with the other treatments. The values of moisture of diets incorporating olive cakes were slightly lower than those found for Celta and its crosses with Duroc and Landrace (70–72%) [39], as well as for Alentejano and Bísaro pigs [38]. The heme pigment contents between 0.50 and 0.90 mg/g were close to those reported for Bísaro (0.45) and Alentejano breeds (0.86) [38]. The ash content was within the values found for the Preto Alentejano and Large White breeds [57]. No significant differences were found between treatments for collagen content, ranging from 1 to 1.5%, a value that was also recorded in a native Croatian breed [58]. The protein content around 23% did not show significant differences between treatments which was also recorded in other pig breeds fed with diets without olive cakes [38,39,57], but also in Pietrain pigs fed with a diet with added olive cake [5]. However, the values obtained in this study for the protein content were 3% higher than those of Korean breeds and crosses with Duroc [55].

### 3.3. Fatty Acid Profile and Lipidic Quality

The fatty acid profile of the LTL muscle is shown in Table 5. No significant differences were found between treatments for total saturated fatty acids (SFAs), monounsaturated fatty acids (MUFAs) and polyunsaturated fatty acids (PUFAs). The predominant SFAs were palmitic acid (C16:0) and stearic acid (C18:0). Although not significant, the proportions of C16:0 and C18:0 varied between the T2 treatment (27.44 and 13.48% for palmitic and stearic, respectively) and the T5 treatment (25.70 and 11.93% for palmitic and stearic, respectively) and, consequently, SFA tended to be higher in the T2 treatment in comparison with treatment T5. This tendency was also observed in Pietrain pigs [5], but they recorded statistically significantly lower values of SFA (38.71% for inclusion of 50 g/kg of olive cake and 34.62% for inclusion of 100 g/kg of olive cake). A similar SFA content was reported for the Alentejano pig [59].

The intramuscular MUFA content of LTL varied between 50.04% for the T2 treatment and 52.80% for treatment T3 and was within the values reported by other authors with the inclusion of 100 g/kg of olive cake [5] but lower than the 55.1% for the Swallow-bellied Mangalitsa and the 58% for the White Mangalitsa found for these two Serbia genotypes recognized as fatty pig breeds [41]. The MUFA content was within the values reported for different local breeds such as the Preto Alentejano breed [57], Iberian pigs [60,61], Mora Romagnola and Casertana breeds [62], Sicilian Black pig [63], Croatian native pig [58], Chinese Heigai pig [64] and Celta pigs [39]. This proportion of MUFA appears to be a trend in the autochthonous southern European swine breeds. However, lower MUFA values were found in the Chinese Ningxiang pig breed [65], Zlotnicka Spotted pigs [66] and Alentejano pig [59]. Regarding the PUFA content, the values varied from 6.49% in treatment T4 to 7.91% in treatment T5. PUFA content depends on the amount and structure of dietary fat, fatty acid synthesis, the rate of conversion to other fatty acids and metabolites and the ratio of oxidation to energy consumption [5]. The inclusion of olive cake in the diet of fed animals (rich in MUFA) reduces the PUFA content in red blood cell membranes [5,67]. The PUFA content for Bísaro pigs recorded is within the reported values for native Croatian pigs [58], Swallow-Bellied Mangalitsa [41], Chinese Ningxiang pigs [65], Zlotnicka Spotted pigs [58] and Celta pigs [39]. However, higher PUFA contents were observed in Pietrain pigs fed a diet with the inclusion of olive cakes (6 to 7% more) [5], Chinese Heigai pigs (4 to 5% more) [64], Iberian pigs (5 to 6% more) [61] Prestice Black-Pied (6 to 7% more) [68] and Alentejano pigs (8 to 9% more) [59].

The PUFA/SFA ratio observed (0.16–0.20) is below the recommended value of 0.4 by the Department of Health and Social Security [69]. The result obtained was close to that found for the Swallow-Bellied Mangalitsa breed [41] and over the 0.15 of native Croatian pigs [58]. However, higher ratios were observed in Alentejano pigs (0.39) [59], Celta pigs (0.38) [39], Heigai pigs (0.38) [64] and Prestice Black-Pied (0.35) [68]. Intermediate values of 0.29 were found in Zlotnicka Spotted pigs [66], and Ningxiang pigs [65]. The content of n-3 and n-6 polyunsaturated fatty acids (PUFAs) in the diet can affect the meat quality, especially in lean pig breeds, and is linked to human health [64]. A low n-6/n-3 PUFA ratio is associated with benefits to human health [70]. The n-6/n-3 PUFA ratio was not significant between treatments. However, the values were very high in relation to the recommended value of 4 [69]. According to several authors [39,67], it is difficult to reduce this value in pork due to the high content of C18:2n-6 present in normal feed concentrate. Therefore, in the present study with diets supplemented with olive cakes with a high content of C18:2n-6 (35–47%; see Table 1), it is not surprising to find a high n-6/n-3 PUFA ratio in the meat. A value of 27.9, close to those recorded by us, is reported for the Alentejano breed [59]. For Serbian genotypes, the White Mangalitsa obtained a much higher ratio of 34% [41] than those obtained in this work as well as in Nero Siciliano pigs (33–37%) [70]. However, lower values were found in other breeds such as 11.2 in the Prestice Black-Pied [68], 7.5 to 10.8 in the Nero Siciliano breed [63], 17 to 18 in Celta pigs [39] or 8.7 in Heigai pigs [64].

The index of atherogenicity (IA) and the index of thrombogenicity (IT) characterize, respectively, the atherogenic and thrombogenic potentials of fatty acids [36]. A fatty acid composition with lower IA and IT values has a better nutritional quality, and its consumption may reduce the risk of coronary heart disease, but no organization has yet provided the recommended values for the IA and IT [71]. However, the cited authors stated that an IA index ranging from 0.16 to 1.3 would characterize the atherogenic potential of fatty acids in meat. No significant differences were observed in the IA and IT in the intramuscular fat of LTL muscle for all treatments and the atherogenicity potential was within the expected range. The IT varied between 0.5 and 0.6. The values for the IA and IT were close to those obtained in Pietrain pigs with the inclusion of 50 and 100 g/kg of olive cake in the diet [5]. The h/H ratio is based on the functional effects of fatty acids on cholesterol metabolism and the higher the h/H ratio, the more nutritionally adequate the oil or fat in the food [72]. The h/H ratio found by us varied between 1.93 and 2.09 and no significant differences were recorded between treatments.

The fatty acid profile of the backfat of the Bísaro pig breed is shown in Table 6. The knowledge of the fatty acid profile of the backfat is very important as this is an important fat source used in the processing of several meat products besides its culinary use. In all treatments, palmitic acid (C16:0), oleic acid (C18:1n-9) and linoleic acid (C18:2n-6) were the most common SFA, MUFA and PUFA present, respectively.

Significant differences (*p* < 0.05) between treatments were found in palmitoleic acid (C16:1n-7), elaidic acid (C17:1n-7) and eicosenoic acid (C20:1n-9). Although there are trace fat components of backfat, the T3 diet shows a significantly higher content of elaidic acid and eicosenoic acids while the T5 diet had a higher content of palmitoleic acid than the other diets. Compared with intramuscular fat of the LTL muscle, backfat did not show the same trends regarding SFA and MUFA. The SFA content varied between 38.69% (T3) and 40.60 (T4). MUFA content ranged from 48.38% (T4) to 50.10% (T3) and the PUFA content from 11.0% (T1) to 11.50% (T5). The results obtained agree with other studies in which olive cake was also added as an ingredient of the diet [5]. The mean value of SFA observed was close to that obtained in the backfat of Prestice Black-Pied [68], Nero Siciliano [63] and Pietrain pigs with the inclusion of 50 and 100 g/kg of olive cake [5] and lower compared to Celta pigs [61] and crossbred barrows (Duroc × Landrace × Large) [42]. The mean value of MUFA obtained in this work was 5 to 6% higher than that obtained by other authors [5,42]. Similar values were also obtained for other breeds such as Nero Siciliano [63] and Prestice Black-Pied [68]. Regarding the PUFA content of backfat, values close to those reported by us were also found by other authors [63,68]. Liotta et al. [5] observed a higher PUFA content in the backfat of Pietrain pigs and its increase was significantly higher with more olive cake introduced into the diet. Globally, the results showed a similarity of SFA and MUFA content between Bísaro and other local pork breeds of south Europe in comparison with other selected and improved breeds such as Pietrain.

No significant differences were found in the PUFA/SFA ratio in the backfat of the Bísaro pork breed. The highest value found, 0.29, was common to three of the treatments (T2, T3 and T5). The control and T4 treatment obtained the same mean value of 0.27 for this ratio. Taking into account the recommended value of 0.4 by the Department of Health and Social Security [69], the PUFA/SFA ratio recorded for Bísaro backfat was better than those described by other authors in other breeds of pigs without the addition of olive cake (0.20 and 0.25) [41] and was also better than the value observed in the intramuscular fat of the LTL muscle (see Table 5). Nevertheless, more favorable values for this ratio were observed in Celta pigs (0.35) and Prestice Black-Pied (0.48) [68]. The n-6/n-3 ratio was identical in all treatments and varied between 20 and 22, within the value in the Mangalitsa breed [41] but higher than those reported in Prestice Black-Pied [68]. A healthy animal product can be characterized by low IA and IT and a high h/H index [72]. As there are no significant differences between treatments, we can point out that the Bísaro pork backfat presented a low saturated fatty acid content, high level of MUFA and adequate IA, IT and h/H indexes according to the lipid quality observed in other breeds [5,73].

As has been mentioned above for the PUFA/SFA ratio, the h/H ratio of backfat was more nutritionally adequate than that seen in the loin. Cavas et al. [72] and Lorenzo et al. [74] suggest that these differences can be caused by higher metabolic activity in intramuscular fat than backfat and different functions of neutral lipids and phospholipids contained in fat from different parts.

## 4. Conclusions

Considering all data obtained in the present research, the main conclusion of this study was that the inclusion of different olive cakes in the diet of Bísaro pigs did not affect the carcass characteristics or the meat and fat quality. In the proportion of 10%, the olive cake can be used as another ingredient in the diet, valorizing a by-product of the olive industry and reducing the environmental impact of olive-mill wastewaters from the extractive industries.

Future studies should be considered, namely the physicochemical characterization of the other muscles and processed meat products and their quality. In addition, some physiological studies on the digestive utilization of these by-products with different added percentages and the nutritional behavior of the animals should be conducted.

## Figures and Tables

**Figure 1 foods-11-01650-f001:**
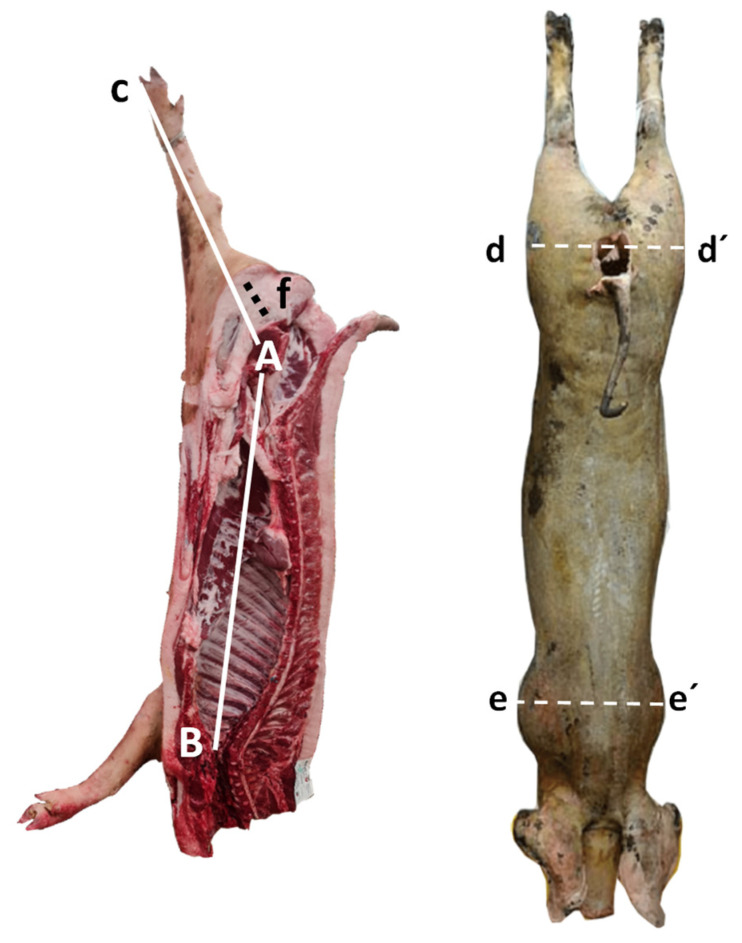
Schematic midsagittal and dorsal view of carcass measurements. (Midsagittal view) carcass length (L), A–B distance; leg length (LL), C–A distance. (Dorsal view) width of the buttocks d–d´ distance; width of the thorax, e–e′ distance; leg fat thickness (f).

**Figure 2 foods-11-01650-f002:**
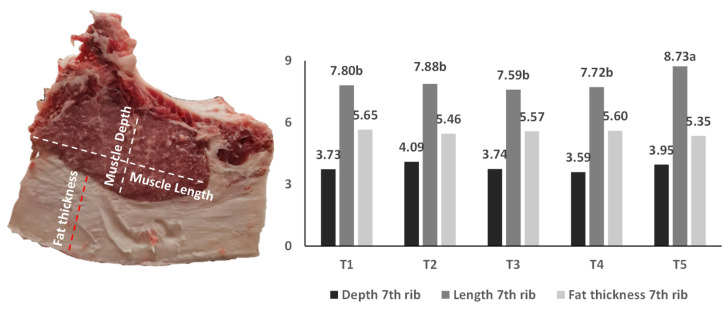
Measurements on *Longissimus thoracis et lumborum* muscle the of Bísaro carcass at the 7th ribs of the loin joint. Mean values with different lowercase letters within the same treatment are significantly different (*p* < 0.05). Mean values without letters have no significant differences (*p* > 0.05). T1—Basic diet and commercial feed; T2—Basic diet + 10% crude olive cake; T3—Basic diet + 10% olive cake, two phases; T4—Basic diet + 10% exhausted olive cake; T5—Basic diet + 10% exhausted olive cake + 1% olive oil.

**Figure 3 foods-11-01650-f003:**
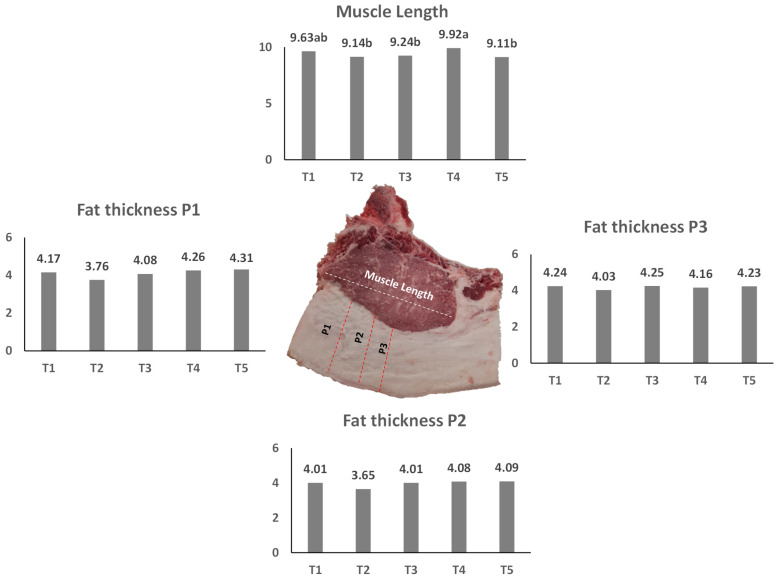
Muscle and backfat measurements at the 13th and 14th rib of Bísaro carcass (P1—4.5 cm from the dorsal midline; P2—6.5 cm from the dorsal midline; P3—8 cm from the dorsal midline). Mean values with different lowercase letters within the same treatment are significantly different (*p* < 0.05). Mean values without letters have no significant differences (*p* > 0.05). T1—Basic diet and commercial feed; T2—Basic diet + 10% crude olive cake; T3—Basic diet + 10% olive cake, two phases; T4—Basic diet + 10% exhausted olive cake; T5—Basic diet + 10% exhausted olive cake + 1% olive oil.

**Figure 4 foods-11-01650-f004:**
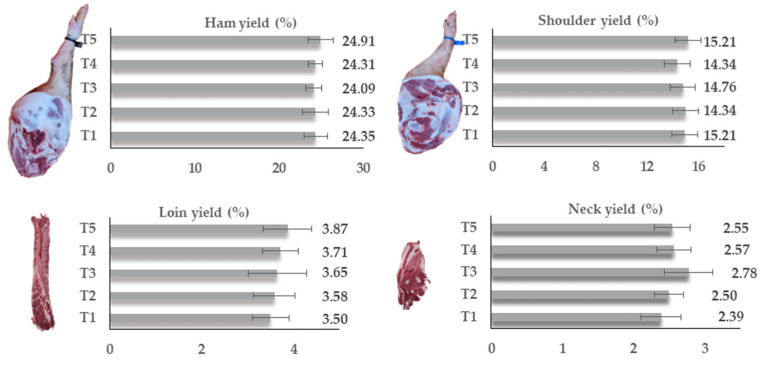
Yield (%) of carcass joints: ham, shoulder, loin and neck. Mean values without letters have no significant differences (*p* > 0.05). T1—Basic diet and commercial feed; T2—Basic diet + 10% crude olive cake; T3—Basic diet + 10% olive cake, two phases; T4—Basic diet + 10% exhausted olive cake; T5—Basic diet + 10% exhausted olive cake + 1% olive oil.

**Table 1 foods-11-01650-t001:** Ingredient composition of the experimental diets (g/kg, as fed basis) and fatty acid composition (g/100 g).

			Diets		
	T1	T2	T3	T4	T5
Olive cake	0	10	10	10	10
Olive oil	0	0	0	0	1
Barley grain	45.80	41.20	41.20	41.20	41.20
Wheat grain	22.60	20.40	20.40	20.40	20.40
Soybean meal 47	12.90	11.60	11.60	11.60	11.60
Rice bran	5.00	4.50	4.50	4.50	4.50
Corn grain	2.50	2.20	2.20	2.20	2.20
DDG corn	5.00	4.50	4.50	4.50	4.50
Beet molasses	4.00	3.60	3.60	3.60	3.60
Minerals and vitamins	1.70	1.70	1.70	1.70	1.70
Supplement min + vit + phytase	0.50	0.30	0.30	0.30	0.30
Chemical composition of diet					
DM	98.05	98.49	98.19	98.15	98.46
OM	93.90	94.20	93.75	94.16	93.98
NDF	18.01	23.39	22.97	24.04	22.88
ADF	6.40	10.62	10.48	10.50	10.06
ADL	0.89	3.06	2.81	3.09	2.86
Cellulose	5.51	7.56	7.68	7.41	7.20
PB	16.00	13.38	13.45	14.39	13.98
GB	5.41	5.53	4.96	4.30	5.20
Fatty acids (g/100 g)					
C13:0	0.19	0.33	0.26	0.28	0
C14:0	0.18	0.13	0.14	0.18	0.14
C16:0	17.60	15.60	15.94	17.28	16.75
C16:1n-7	0.11	0.22	0.18	0.13	0.23
C18:0	2.06	2.54	2.32	2.09	2.88
C18:1n-9	25.85	39.93	36.05	27.37	33.79
C18:1n-7	0.99	1.32	1.17	0.96	1.17
C18:2n-6	47.55	35.31	39.06	46.34	40.76
C18:3n-3	2.64	2.05	2.28	2.64	2.33
C20:0	0.48	0.48	0.48	0.47	0.47
C20:1n-9	0.45	0.40	0.42	0.45	0.41
C22:0	0.29	0.27	0.27	0.29	0.27
C22:1n-9	0.48	0.37	0.37	0.43	0.39
C24:0	0.33	0.29	0.29	0.34	0.29
ΣSFA	21.54	20.03	20.09	21.31	20.57
ΣMUFA	28.00	42.36	38.31	29.45	36.10
ΣPUFA	50.46	37.61	41.60	49.24	43.34
PUFA/SFA	2.34	2.07	1.88	2.31	2.10
n-6/n-3	17.58	16.66	16.69	17.10	16.90

DM—dry matter; OM—organic matter; NDF—neutral detergent fiber; ADF—acid detergent fiber; ADL—acid detergent lignin; PB—crude protein; GB—crude fat; SFA, Saturated fatty acid; MUFA, Monounsaturated fatty acid; PUFA, Polyunsaturated fatty acid; the n-6/n-3 (∑ omega-6) (∑ omega-3). T1—Basic diet and commercial feed; T2—Basic diet + 10% crude olive cake; T3—Basic diet + 10% olive cake, two phases; T4—Basic diet + 10% exhausted olive cake; T5—Basic diet + 10% exhausted olive cake + 1% olive oil.

**Table 2 foods-11-01650-t002:** Carcass conformation measurements (cm). Effect of treatment with olive cake.

Carcass Data (cm)	T1	T2	T3	T4	T5	SE	Significance
Width of the buttocks (d–d′)	36.45	34.63	36.24	34.81	36.06	1.19	ns
Width of the thorax (e–e′)	35.78	35.65	35.65	34.98	35.59	1.10	ns
Carcass length (L)	90.70	91.11	90.24	92.00	90.66	1.61	ns
Carcass depth	41.69	43.15	41.30	43.63	42.55	0.95	ns
Leg measurements (cm)							
Leg length (LL)	65.18	65.49	60.88	65.61	64.43	1.44	ns
Leg fat thickness	8.45	8.48	7.64	8.66	8.00	0.37	ns
Backfat thickness (cm)							
1st cervical	5.70	5.96	5.78	5.38	5.88	0.61	ns
4th/5th cervical	6.50	7.46	6.58	6.39	5.93	0.52	ns
2nd/3rd lumbar	4.58	4.39	4.68	4.57	4.66	0.33	ns
4th/5th lumbar	4.85	4.90	4.66	4.98	5.17	0.46	ns

ns—Not significant; SE—Standard error; T1—Basic diet and commercial feed; T2—Basic diet + 10% crude olive cake; T3—Basic diet + 10% olive cake, two phases; T4—Basic diet + 10% exhausted olive cake; T5—Basic diet + 10% exhausted olive cake + 1% olive oil.

**Table 3 foods-11-01650-t003:** Effect of different olive cake treatments on carcass and physical meat characteristics of Bísaro pigs.

Carcass Data	T1	T2	T3	T4	T5	SE	Significance
BW	139.20	143.06	134.00	141.95	140.55	4.52	ns
CW (kg)	109.59	111.68	105.84	111.01	111.05	3.79	ns
Dressing percentage (%)	78.73	78.06	78.98	78.20	79.01	2.68	ns
LCSW (kg)	54.25	55.44	52.40	54.85	54.77	1.78	ns
WHC (%)	12.44ab	13.04ab	18.45a	12.11b	12.48ab	2.19	*
SF (Kgf)	3.70	4.17	3.90	3.41	3.76	0.40	ns
pH (1 h)	6.37	6.22	6.34	6.40	6.26	0.08	ns
pH (24 h)	5.71	5.76	5.74	5.68	5.73	0.06	ns
Color parameters							
*L**	52.02	53.43	51.77	51.98	53.24	1.65	ns
*a**	13.85	13.93	13.39	11.72	13.18	0.63	ns
*b**	11.48	12.15	11.30	10.26	11.34	0.62	ns
H*	39.34	41.08	40.16	41.19	40.81	1.03	ns
C*	18.01	18.51	17.54	15.60	17.40	0.82	ns

ns—Not significant (*p* ≥ 0.05); * *p* < 0.05; Mean values with different lowercase letters within the same treatment are significantly different (*p* < 0.05). Mean values without letters have no significant differences (*p* > 0.05). SE—Standard error; BW—Live weight; CW—Carcass weight; LCSW—Left side carcass weight; WHC—Water holding capacity (%); SF—Shear force (Kgf); (*L**) luminosity index; (*a**) red index; (*b**) yellow index; (C*) chroma and (H*) tom. T1—Basic diet and commercial feed; T2—Basic diet + 10% crude olive cake; T3—Basic diet + 10% olive cake, two phases; T4—Basic diet + 10% exhausted olive cake; T5—Basic diet + 10% exhausted olive cake + 1% olive oil.

**Table 4 foods-11-01650-t004:** Chemical composition of *Longissimus thoracis et lumborum* muscle. Effect of treatment with olive cake.

	T1	T2	T3	T4	T5	SE	Significance
Total fat (%)	4.86b	7.22a	5.54ab	5.60ab	6.52ab	0.66	*
Ash (%)	1.44	1.35	1.50	1.42	1.36	0.07	ns
Moisture (%)	70.10a	67.55b	68.98ab	69.51ab	69.04ab	0.79	*
Heme pigments (mg/g)	0.57	0.90	0.50	0.60	0.76	0.28	ns
Collagen (%)	1.15	1.41	1.20	1.44	1.46	0.16	ns
Protein (%)	22.68	22.40	22.88	22.45	22.80	0.38	ns

ns—Not significant, * *p* < 0.05; Mean values with different lowercase letters within the same treatment are significantly different (*p* < 0.05). Mean values without letters have no significant differences (*p* > 0.05). SE—Standard error. Heme pigments in mg myoglobin/g fresh muscle. T1—Basic diet and commercial feed; T2—Basic diet + 10% crude olive cake; T3—Basic diet + 10% olive cake, two phases; T4—Basic diet + 10% exhausted olive cake; T5—Basic diet + 10% exhausted olive cake + 1% olive oil.

**Table 5 foods-11-01650-t005:** Fatty acid profile of intramuscular fat of the *Longissimus thoracis et lumborum* muscle from the Bísaro pig breed. Effect of treatment with olive cake.

				Diets			
Fatty Acids	T1	T2	T3	T4	T5	SE	Significance
C10:0	0.05	0.05	0.06	0.06a	0.22	0.07	ns
C14:0	1.23	1.25	1.27	1.25a	1.19	0.04	ns
C15:0	0.01	0.07	0.12	0.04a	0.12	0.03	ns
C16:0	26.35	27.44	26.26	26.77a	25.70	0.72	ns
C16:1n-7	3.13	3.07	3.20	3.31a	2.93	0.16	ns
C17:0	0.15	0.16	0.17	0.17a	0.18	0.01	ns
C17:1n-7	0.17	0.17	0.18	0.18a	0.19	0.01	ns
C18:0	12.38	13.48	12.22	12.30a	11.93	0.41	ns
9t-C18:1	0.14	0.13	0.12	0.15a	0.13	0.01	ns
C18:1n-9	48.28	45.87	48.47	44.82a	48.65	1.42	ns
C18:2n-6	5.88	6.47	5.92	5.53a	6.96	0.47	ns
C20:1n-9	0.77	0.76	0.74	0.74a	0.72	0.03	ns
C18:3n-3	0.19	0.23	0.19	0.21a	0.25	0.19	ns
C20:2n-6	0.22	0.24	0.19	0.21a	0.25	0.19	ns
C20:4n-6	0.42	0.25	0.40	0.34a	0.34	0.10	ns
ΣSFA	40.52	42.67	40.33	40.88a	39.41	1.10	ns
ΣMUFA	52.57	50.04	52.80	52.63a	52.68	1.24	ns
ΣPUFA	6.91	7.29	6.87	6.49	7.91	0.51	ns
PUFA/SFA	0.17	0.17	0.17	0.16	0.20	0.01	ns
n-6/n-3	25.21	29.93	26.78	21.12	26.25	2.59	ns
IA index	0.53	0.62	0.53	0.54	0.50	0.04	ns
IT index	1.31	1.55	1.31	1.33	1.25	0.10	ns
h/H	1.99	1.93	2.01	1.94	2.09	0.06	ns

ns—Not significant, * *p* < 0.05; Mean values with different lowercase letters within the same treatment are significantly different (*p* < 0.05). Mean values without letters have no significant differences (*p* > 0.05). SE—Standard error; SFA, Saturated fatty acid; MUFA, Monounsaturated fatty acid; PUFA, Polyunsaturated fatty acid; n-6/n-3 (∑ omega-6) (∑ omega-3); IA, Index of atherogenecity; IT, Index of thrombogenicity; h/H, Hypocholesterolemic/hypercholesterolemic index. Only fatty acids which represented more than 0.1% are presented in the table, although all detected fatty acids were used for calculating the totals and the indices. T1—Basic diet and commercial feed; T2—Basic diet + 10% crude olive cake; T3—Basic diet + 10% olive cake, two phases; T4—Basic diet + 10% exhausted olive cake; T5—Basic diet + 10% exhausted olive cake + 1% olive oil.

**Table 6 foods-11-01650-t006:** Fatty acid profile of backfat from the Bísaro pig breed. Effect of treatment with olive cake.

				Diets			
Fatty Acids	T1	T2	T3	T4	T5	SE	Significance
C14:0	1.12	1.06	1.14	1.16	1.04	0.04	ns
C16:0	25.58	24.87	24.72	25.95	24.82	0.46	ns
C16:1n-7	1.68ab	1.49b	1.67ab	1.79a	1.52b	0.07	*
C17:0	0.22	0.24	0.27	0.24	0.25	0.01	ns
C17:1n-7	0.19a	0.20a	0.24a	0.21ab	0.21ab	0.01	*
C18:0	13.23	13.36	12.21	12.89	12.93	0.35	ns
9t-C18:1	0.17	0.16	0.16	0.14	0.14	0.02	ns
C18:1n-9	45.35	45.93	46.89	45.21	46.15	0.66	ns
C18:2n-6	9.83	10.29	9.96	9.8	10.30a	0.31	ns
C20:0	0.23	0.20	0.21	0.22	0.22	0.01	ns
C20:1n-9	1.04a	0.87b	1.08a	0.96ab	1.02a	0.05	*
C18:3n-3	0.38	0.41	0.40	0.39	0.41	0.01	ns
C20:2n-6	0.50	0.49	0.54	0.48	0.51	0.02	ns
C20:3n-3	0.09	0.07	0.10	0.09	0.08	0.01	ns
C20:4n-6	0.12	0.12	0.12	0.12	0.13	0.01	ns
ΣSFA	40.51	39.83	38.69	40.60	39.40	0.73	ns
ΣMUFA	48.49	48.71	50.11	48.38	49.10	0.67	ns
ΣPUFA	11.00	11.46	11.20	11.02	11.50	0.35	ns
PUFA/SFA	0.27	0.29	0.29	0.27	0.29	0.01	ns
n-6/n-3	22.00	22.19	20.75	21.15	21.84	0.53	ns
IA index	0.51	0.49	0.48	0.52	0.48	0.01	ns
IT index	1.29	1.25	1.19	1.29	1.23	0.04	ns
h/H	2.09	2.20	2.23	2.05	2.22	0.07	ns

ns—Not significant, * *p* < 0.05; Mean values with different lowercase letters within the same treatment are significantly different (*p* < 0.05). Mean values without letters have no significant differences (*p* > 0.05). SE—Standard error. SFA—Saturated fatty acid; MUFA—Monounsaturated fatty acid; PUFA—Polyunsaturated fatty acid; n-6/n-3 (∑ omega-6) (∑ omega-3); IA—Index of atherogenecity; IT—Index of thrombogenicity; h/H—Hypocholesterolemic/hypercholesterolemic index. Only fatty acids which represented more than 0.1% are presented in the table, although all detected fatty acids were used for calculating the totals and the indices. T1—Basic diet and commercial feed; T2—Basic diet + 10% crude olive cake; T3—Basic diet + 10% olive cake, two phases; T4—Basic diet + 10% exhausted olive cake; T5—Basic diet + 10% exhausted olive cake + 1% olive oil.

## Data Availability

Data is contained within the article.

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
