# Peer review of "Can the Introduction of Different Olive Cakes Affect the Carcass, Meat and Fat Quality of Bísaro Pork?"

_foods, 2022, doi:10.3390/foods11111650_

Round 1

Reviewer 1 Report

General comments:

The manuscript deals with the evaluation of alternative feed ingredients (olive by products) for feeding pigs and the effect of their dietary inclusion on carcass, meat and fat quality. In my opinion, the topic addressed in presented manuscript is highly relevant for animal based food production and falls into the scope of journal “Foods”. Namely, recently a lot of attention is given to replacement of feed ingredients, especially soybean and corn, with cheaper agricultural by products of local origin.

In general the manuscript is well written and organised. However, several information are missing in material and method section and in description of figures. Also I suggest that the results are simplified and English editing of the manuscript.

I recommend a minor revision of the manuscript in present form and suggest the following changes.

Specific comments:

Introduction

Line 76-77: These are positive effects if row meat is consumed/the final product. However, if meat is processed to dry-cured products more saturated fat is desired – please address this in the introduction.

Abstract:

Line 25-28: I suggest changing the sentence to: “The present work aims to evaluate the inclusion of different olive cakes in the feed of Bisaro pigs. The effect on carcass traits, meat quality and lipid composition of meat and backfat were evaluated after feeding commercial diet supplemented with 10% of either crude olive cake, exhausted olive cake without or with 1% of olive oil, or two-phases olive cake for 90 days.”

Line 28-28: I suggest deleting “(1 and 24 h after slaughter)”

Line 30: replace “of” with “at”

Line 31: replace “ribs” with “rib” and “pork” with “pig”

Line 34: I suggest “confirm normal carcass without quality deviations as DFD or PSE:”

Line 35: replace “firmness” with “firm” and delete “to”

Line 37: replace ”in the carcass” with “on the carcass”

Materials and methods

Several information are missing in this section:

  • fattening system should be briefly described (individual or group housing, season, n of animals per pen,….)
  • feeding level (consumption per pig per day), was it on ad libitum basis or restricted?
  • incorporation of by-products in the diet should be described (was it included into the diet – mixed in the course of diet preparation or fed additionally to the diet?)
  • include information on basic chemical composition of experimental diets and ME or NE (in addition to ingredient and fatty acid composition given in Table 1).
  • I suggest to report average backfat thickness (average of P1. P2 and P3) or you chose 1 to present.
  • describe the procedure of sampling and measurements of backfat and muscle thickness (presented on Fig. 3 and 4) – you only present carcass measurments with metal tape

Line 87: replace “in relation a basic” with “ in combination with a”

Line 106: I suggest “Pigs were slaughtered at an average body weight of 135± SD kg in the …”

Line 118-122: I assume that the meat colour was measured on LD sample not on whole carcass and this section should be moved to section 2.4?

Table 1: I suggest changing to “Ingredient (g/kg, as fed) and fatty acid composition (g/100 g fat)”

Figure 2: I suggest “Schematic lateral view of mid-carcass (left picture) and dorsal view of carcass (right picture): ….”

Line 156 and 199: replace “joints” with “cuts”

Results and discussion

I encourage authors to separate results and discussion.

When comparing results to previous studies please include the information on how long olive by products were fed in these studies (e.g. in line 204)

Line 199: please replace ”joints” with “cuts”

Line 219: include “yield of” before “most important”

Line 223: I assume that “muscle depth” should be added before “13th”

Figure 3: I suggest to change the figure description to “Measurements on the loin cross-cut of Bisaro pigs (fed diet supplemented with 10% of different olive cakes for 90 days) at the level of the 7th rib (cm; p<0.05)”, also superscripts “a” and “b” should be given only to the traits which are significantly different – the same should be applied to Figure 4

Figure 4 is hard to read and the measurements of backfat at different positions (P1, P2, P3) should be averaged and presented as the average of backfat thickness on a cross-cut of loin at the position between 13th and 14th thoracic vertebrae unless authors present a reason why 3 repetitions are presented. In my opinion on this pictures loin eye area (cm2) and fat over loin eye (cm2) would by more informative as this is a standardised measurement.

Table 2 -4: please delete superscripts “a”in traits where comparison between treatment groups is non-significant, lase add description of leg fat thickness (I can find it in materials and methods), add the number of samples per measurement/trait (n)

Line 247: I suggest “In the Table 2 measurements of carcass conformation are shown”

Line 248: please add “for the” before “leg”

Line 293: more reddish colour of the meat develops with age, which is much more relevant then BW – as traditional fatty type pigs grow more slowly they reach target slaughter weight at an older age- and meat more mature. Other factor to consider when comparing results is the rearing system- as outdoor system increases aerobic capacity of glycolytic muscles due to additional exercise.

Line 403: please change to “Intramuscular fat MUFA content of Longissimus…”

Author Response

Dear review,

All modifications were made following the reviewer's suggestions and comments, and responses to their comments are also attached. Thanks to their recommendations, significant modifications were made throughout the manuscript.

Thank you for your attention.

Answers to Reviewer 1

General comments:

The manuscript deals with the evaluation of alternative feed ingredients (olive by products) for feeding pigs and the effect of their dietary inclusion on carcass, meat and fat quality. In my opinion, the topic addressed in presented manuscript is highly relevant for animal based food production and falls into the scope of journal “Foods”. Namely, recently a lot of attention is given to replacement of feed ingredients, especially soybean and corn, with cheaper agricultural by products of local origin. In general the manuscript is well written and organised. However, several information are missing in material and method section and in description of figures. Also I suggest that the results are simplified and English editing of the manuscript. I recommend a minor revision of the manuscript in present form and suggest the following changes.

Introduction

Line 76-77: These are positive effects if row meat is consumed/the final product. However, if meat is processed to dry-cured products more saturated fat is desired – please address this in the introduction

Response: Suggestion accepted; changes have been made in the revised version of the manuscript.

Abstract:

Line 25-28: I suggest changing the sentence to: “The present work aims to evaluate the inclusion of different olive cakes in the feed of Bisaro pigs. The effect on carcass traits, meat quality and lipid composition of meat and backfat were evaluated after feeding commercial with 10% of either crude olive cake, exhausted olive cake without or with 1% of olive oil, or twophases olive cake for 90 days.”

Response: Suggestion accepted; changes have been made in the revised version of the manuscript.

Line 28-28: I suggest deleting “(1 and 24 h after slaughter)”

Response: Suggestion accepted; changes have been made in the revised version of the manuscript.

Line 30: replace “of” with “at”

Response: Suggestion accepted; changes have been made in the revised version of the manuscript.

Line 31: replace “ribs” with “rib” and “pork” with “pig”

Response: Suggestion accepted; changes have been made in the revised version of the manuscript.

Line 34: I suggest “confirm normal carcass without quality deviations as DFD or PSE:”

Response: Suggestion accepted; changes have been made in the revised version of the manuscript.

Line 35: replace “firmness” with “firm” and delete “to”

Response: Suggestion accepted; changes have been made in the revised version of the manuscript.

Line 37: replace ”in the carcass” with “on the carcass”

Response: Suggestion accepted; changes have been made in the revised version of the manuscript.

Materials and methods

Several information are missing in this section: fattening system should be briefly described (individual or group housing, season, n of animals per pen,….) feeding level (consumption per pig per day), was it on ad libitum basis or restricted? incorporation of by-products in the diet should be described (was it included into the diet – mixed in the course of diet preparation or fed additionally to the diet?) include information on basic chemical composition of experimental diets and ME or NE (in addition to ingredient and fatty acid composition given in Table 1).

Response: Suggestion accepted; changes have been made in the revised version of the manuscript.

I suggest to report average backfat thickness (average of P1. P2 and P3) or you chose 1 to present. describe the procedure of sampling and measurements of backfat and muscle thickness (presented on Fig. 3 and 4) – you only present carcass measurements with metal tape

Response: Suggestion accepted; changes have been made in the revised version of the manuscript.

Line 87: replace “in relation a basic” with “ in combination with a”

Response: Suggestion accepted; changes have been made in the revised version of the manuscript.

Line 106: I suggest “Pigs were slaughtered at an average body weight of 135± SD kg in the …”

Response: Suggestion accepted; changes have been made in the revised version of the manuscript.

Line 118-122: I assume that the meat colour was measured on LD sample not on whole carcass and this section should be moved to section 2.4? Table 1: I suggest changing to “Ingredient (g/kg, as fed) and fatty acid composition (g/100 g fat)” Figure 2: I suggest “Schematic lateral view of mid-carcass (left picture) and dorsal view of carcass (right picture): ….”

Response: Suggestion accepted; changes have been made in the revised version of the manuscript.

Line 156 and 199: replace “joints” with “cuts”

Response: Suggestion accepted; changes have been made in the revised version of the manuscript.

Results and discussion

I encourage authors to separate results and discussion. When comparing results to previous studies please include the information on how long olive by products were fed in these studies (e.g. in line 204)

Response: We appreciate the suggestion; however, we consider that the results and the discussion should be considered at the same time.

Line 199: please replace ”joints” with “cuts”

Response: Suggestion accepted; changes have been made in the revised version of the manuscript.

Line 219: include “yield of” before “most important”

Response: Suggestion accepted; changes have been made in the revised version of the manuscript.

Line 223: I assume that “muscle depth” should be added before “13 Figure 3: I suggest to change the figure description to “Measurements on the loin cross-cut of Bisaro pigs (fed diet supplemented with 10% of different olive cakes for 90 days) at the level of the 7 rib (cm; p<0.05)”, also superscripts “a” and “b” should be given only t the traits which are significantly different – the same should be applied to Figure 4.

Response: Suggestion accepted; changes have been made in the revised version of the manuscript.

Figure 4 is hard to read and the measurements of backfat at different positions (P1, P2, P3) should be averaged and presented as the average of backfat thickness on a crosscut of loin at the position between 13 and 14 thoracic vertebrae unless authors present a reason why 3 repetitions are presented. In my opinion on this pictures loin eye area (cm ) and fat over loin eye (cm ) would by more informative as this is a standardised measurement.

Response: P1, P2, P3 backfat measurements are a standardized distances from the dorsal midline of the vertebrae and provide information about the fat distribution over the LTL muscle. We agree with the muscle area, but it was not measured.

Table 2 -4: please delete superscripts “a”in traits where comparison between treatment groups is non-significant, lase add description of leg fat thickness (I can find it materials and methods), add the number of samples per measurement/trait (n)

Response: Suggestion accepted; changes have been made in the revised version of the manuscript.

Line 247: I suggest “In the Table 2 measurements of carcass conformation are shown”

Response: Suggestion accepted; changes have been made in the revised version of the manuscript.

Line 248: please add “for the” before “leg”

Response: Suggestion accepted; changes have been made in the revised version of the manuscript.

Line 293: more reddish colour of the meat develops with age, which is much more relevant then BW – as traditional fatty type pigs grow more slowly they reach target slaughter weight at an older age- and meat more mature. Other factor to consider when comparing results is the rearing system- as outdoor system increases aerobic capacity of glycolytic muscles due to additional exercise

Response: The text was rewritten according the referee’s suggestion

Line 403: please change to “Intramuscular fat MUFA content of Longissimus…”

Response: Suggestion accepted; changes have been made in the revised version of the manuscript.

Reviewer 2 Report

Manuscript Number: foods-1733234-peer-review-v1

Title: Can the introduction of different olive cakes affect the carcass and meat quality of Bísaro pork?

Authors: Ana Leite, Rubén Domínguez, Lia Vasconcelos, Iasmin Ferreira, Etelvina Pereira, Victor Pinheiro, D. Outor-Monteiro, Sandra Rodrigues, José Manuel Lorenzo, Eva María Santos, Silvina Cecilia Andrés, Paulo C. B. Campagnol, and Alfredo Teixeira

Overview and general recommendation:

The aim of the manuscript was to evaluate the potentiality of inclusion of 10% of different olive cakes in the feeding Bísaro pigs and its effect on carcass traits and meat quality and lipid quality of meat and backfat. Unfortunately, although the research topic is interesting,The article contains numerous insufficiency and ambiguities (in parts: Introduction, Material and methods as well as Results and discussion). Some parts of the Results and discussion chapter have a chaotic and illogical composition. The main objection, however, is the low substantive level of results and discussion. Most of the discussions of the obtained results with the results of the works of other authors are not related to the purpose of the research. The research concerns the influence of pig nutrition on the quality of carcasses and meat, but in the discussion the authors most often refer to the influence of the pig breed on the examined characteristics of carcasses and meat. Numerous other comments are provided below :

Major comments:

There is no logical sequence of the issues discussed in the following parts:

- Lines 28 and 29. The information on pH should be provided together with the information on meat quality (after discussing carcass traits)

- Lines 114-118 Information on performing the measure of the pH and the color of meat should be transferred to the section where the methodology of meat quality assessment is described. (after the description of carcass quality assessment methods). The measurement of the pH and the color of the meat do not match the chapter „Slaughter procedure”

- Line 198 In order to keep the logical structure of the discussed content, it is first necessary to present the results of measurements performed for the whole carcass (weight, length), and only then the shares of the main elements.

Lines 33-34 and 288-289 The manuscript does not explain the classification criteria used by the authors when classifying meat as: “good quality meat” and with PSE or DFD defects (it applies to pH and color parameters). Without giving the classification criteria and performing the classification, it cannot be said that all the tested meat samples were not PSE or DFD.

Lines 38-40. Does the sentence: “In addition, the inclusion of this olive industry by-product in the animal diet also has an important economic implication, since it would make the pig feeding cheaper, especially taking into account the increasingly high prices of the cereals used for the manufacture of feed.” result from the conducted research? Has such an analysis been performed? Was, for example, the effect of feed modification on daily feed consumption and pigs' daily growth rate analyzed? If this does not result from the research carried out, it cannot be written about it in the abstract. Did the authors determine the costs of conventional and modified feed? This comment also applies to: Lines 531-532 the sentence does not result from the conducted research. It is not known what ingredients can be replaced by olive cake and whether it actually reduces costs.

Whole Introduction. Since the authors in the title and for in the aim of research emphasize the breed of pigs - the introduction should contain information why this breed was chosen for research.

Whole Introduction. It should be more emphasized what is new in the article (in relation to other research on the inclusion of olive cakes in pig nutrition).

Line 86. N = 8 for one food group is not enough. With such limited research material, we can rather talk about preliminary research.

Chapter 2.1. It should be clarified whether the animals were of the same age at the beginning of the experiment, whether the animals were fed the same until the beginning of the experiment (i.e. until the mass was 100 kg), if so, how they were fed, whether the animals were kept in the same conditions , in which? If the animals were not initially aligned then the experiment should be considered as incorrectly designed.

Lines 230-231; Lines 238-243; Lines 251-266; Lines 277-284; Lines 289-301; Lines 331-336; many parts of a paragraph about the chemical composition of meat; and Lines 373-380; Lines 407-412 - many fragments with the composition of fatty acids: The discussion has nothing to do with the purpose of the research. It is unnecessary in this form. The discussion should be focused on the purpose of the research.

Minor comments:

Line 2. The title makes reference to the quality of the meat - and there is no reference to the quality of the fat - the title should be reworded

Lines 26-27. The types of feed should be listed in the order in which they appear in the tables (e.g. table 1)

Lines 32-33. This sentence is incomprehensible.

Line38. In order to provide "as a general conclusion" information on the quality of fat in the abstract, information on this subject should be provided earlier in the text.

Lines 41-42. Keywords it is worth expanding with the terms concerning the other tested material properties.

Line 86. The breed of pigs must be indicated.

Line 87. Explain why this particular amount of olive oil cakes was added to the feed.

Lines 87-88. It should be clarified whether the feeds were balanced in terms of the content of basic ingredients (protein, fat, carbohydrates)? What was the content of the basic chemicals in the individual feeds?

Table 1. Why the weight of the feed ingredients does not sum up to 100,00

Lines 89-90. The highest total PUFA value was found in the control feed, why? why do the authors omit it and how can this be explained? Moreover, the differences in the SFA are very small. Generally, due to the small differences in the quoted values - without a statistical analysis, it cannot be said that the values were higher.

Line 114 At what minute after slaughter, the first pH measurement was made (this is important due to the quality of the meat). What muscle was the pH measured in?

Line 118. How long after slaughter was the color measured? What conditions were applied (bloming time?).

Line 119, Line 286. redness (a*), and yellowness (b*) – this is a simplification - redness and yellowness only apply to positive values of these parameters.

Line 137. Are the measurements of the carcass length, limbs etc. are essential in the experiment. Is there still a clear development of the skeletal system at this stage of the pig's life, when the experiment was carried out?

Line 155 – please add „and fat”

Line 163 – What was the size of the heated sample. Was the temperature of the medium the same as the end temperature of the sample? (usually the temperature of the heating medium is a few degrees higher).

Lines 165 -166. What was the time from warming up to measuring the cutting force? What was the temperature of the samples at the time of testing.

Line 192; for *P < 0.05, **P < 0.01 or ***P < 0.001 significance levels – it is unnecessary - the authors only have in the tables / figures *P < 0.05.

Lines 193-194: For the sensorial statistical analysis of the results, an analysis of variance (ANOVA) of one-way using IBM SPSS Statistics 22.0 program (IBM Corporation, Somers, NY, USA) was performed for variables considered in the study…. There is no sensory research in the article.

Line 221. Figure 2 should be supplemented with information on the statistical analysis (symbols in the figures, information in the title of the figure).

Figures 3, 4,. Tabless 2, 3, 4, 5, 6, In the captions, information about the statistical analysis should be provided (what the letter symbols mean), In the captions under the tables, it should be written what SE means.

Figure 2, 3, 4. Axes should start at 0.

Figure 4. The drawing is illegible.

All tables and figures: To make it easier to read the article, it is worth giving an explanation of abbreviations T1, T2, T3, T4, T5 under the individual tables and figures (now they are only in the methodology under table 1).

Figures 3, 4. Why is there no standard deviation in these figures (in Figure 2 there is).

Lines 223 -224 Same sentence twice? Moreover, what does it mean depth at the level of the 7th rib? No detailed explanation in the methodology - should be completed.

Lines 225-226. What does “length of the muscle” mean? No detailed explanation in the methodology - should be completed. Instead of length of the muscle, it is better to measure and discuss the cross-sectional area of the muscle.

Lines 228, 229 The values vary between ... .. it should be added that "mean values"

Line 249 Information on backfat thickness has not been provided in detail in the methodology. In addition, what was the purpose of checking backfat thickness at so many points (in 9) - did the authors expect that a feed modification would change backfat thickness at only one point in the carcass?

Table 3. The table methodology does not specify what Dressing percentage (%) means?

Tables 2-5 The form of the titles should be standardized. Now there are different forms, e.g. Effect of different olive cake treatments on …. or e.g..  Chemical composition … for the 5 diet treatments

Line 348. The title is incorrect WHC and SF it is not a chemical composition.

Lines 323-324: On what basis do the authors claim that “The lower content of oleic acid in the control diet (T1) in comparison with the other treatments (see Table 1) could be the origin of these differences.” Do other authors also point to the influence of this fatty acid?

Lines 352-353: The sentence “It was interesting to note that the effect of the diet treatment on moisture had an inverse trend to intramuscular fat content.” does not indicate an "interesting" relationship, but an obvious relationship between the chemical components in the meat.

Line 372. The T3 had a significantly higher WHC value than T4 – abbreviation T3 and T4 should be replaced with text.

Lines 34-35; 380-381: On what basis do the authors claim that: The values of water holding capacity and shear force observed conform to a non-exudative and firmness meat and The values of WHC and SF observed conform to a non-exudative and firm meat.?

Lines 388-391; 458-461. In my opinion, as the differences are not statistically significant, it is not worth mentioning. The differences the authors write about are rather small - so there is no sense in pointing to the existence of a tendency.

Line 478. In case of margaric acid (C17:1n-7) – all samples are marked with „a”

Lines 523-524. On what basis the authors write such a statement?

Lines 534-548. In my opinion, these sentences are unnecessary.

Author Response

Dear review,

All modifications were made following the reviewer's suggestions and comments, and responses to their comments are also attached. Thanks to their recommendations, significant modifications were made throughout the manuscript.

Thank you for your attention.

Answers to Reviewer 3

The aim of the manuscript was to evaluate the potentiality of inclusion of 10% of different olive cakes in the feeding Bísaro pigs and its effect on carcass traits and meat quality and lipid quality of meat and backfat. Unfortunately, although the research topic is interesting,The article contains numerous insufficiency and ambiguities (in parts: Introduction, Material and methods as well as Results and discussion). Some parts of the Results and discussion chapter have a chaotic and illogical composition. The main objection, however, is the low substantive level of results and discussion. Most of the discussions of the obtained results with the results of the works of other authors are not related to the purpose of the research. The research concerns the influence of pig nutrition on the quality of carcasses and meat, but in the discussion the authors most often refer to the influence of the pig breed on the examined characteristics of carcasses and meat. Numerous other comments are provided below.

Response: The data of the present work are part of the project research “BisOlive: Use of olive pomace in the feeding of Bísaro swine. Evaluation of the effect on meat quality” project. NORTE-01-0247-FEDER-072234. In this manuscript we analyzed the effect of the different olive cakes in the carcass and meat quality of Bísaro pork. Bísaro is a rare pork breed reared in the area where the project was carried out. As general conclusion as we could see from the results is that any olive cake would be used as a feed ingredient, and it will be interesting for olive oil industry finding out a way for this waste. As we decided to submit this manuscript to this special issue on “Improving the Meat and Meat Products Quality of Rare Pork Breeds and Genetic Types” we find natural to give more importance to “the influence of the pig breed on the examined characteristics of carcasses and meat”. However, we have tried to reduce the “chaotic an illogical composition” of results and the discussion thanks to referee comments and suggestions

Major comments:

There is no logical sequence of the issues discussed in the following parts:

Lines 28 and 29. The information on pH should be provided together with the information on meat quality (after discussing carcass traits) –

Response: Suggestion accepted; changes have been made in the revised version of the title manuscript.

Lines 114-118 Information on performing the measure of the pH and the color of meat should be transferred to the section where the methodology of meat quality assessment is described. (after the description of carcass quality assessment methods). The measurement of the pH and the color of the meat do not match the chapter „Slaughter procedure” –

Response: changes have been made in the revised version of the title manuscript (physicochemical analysis of meat ant fat)

Line 198 In order to keep the logical structure of the discussed content, it is first necessary to present the results of measurements performed for the whole carcass (weight, length), and only then the shares of the main elements.

Response: changes have been made in the revised version

Lines 33-34 and 288-289 The manuscript does not explain the classification criteria used by the authors when classifying meat as: “good quality meat” and with PSE or DFD defects (it applies to pH and color parameters). Without giving the classification criteria and performing the classification, it cannot be said that all the tested meat samples were not PSE or DFD.

Response: From the different criteria to classify PSE, DFD carcasses we used the most consensual and objective the pH and the kinetics descending values from slaughter to 24 hours after 4° C freezing and the CIELAB color coordinates.

Lines 38-40. Does the sentence: “In addition, the inclusion of this olive industry by-product in the animal diet also has an important economic implication, since it would make the pig feeding cheaper, especially taking into account the increasingly high prices of the cereals used for the manufacture of feed.” result from the conducted research? Has such an analysis been performed? Was, for example, the effect of feed modification on daily feed consumption and pigs' daily growth rate analyzed? If this does not result from the research carried out, it cannot be written about it in the abstract. Did the authors determine the costs of conventional and modified feed? This comment also applies to:

Response: The text was modified.

Lines 531-532 the sentence does not result from the conducted research. It is not known what ingredients can be replaced by olive cake and whether it actually reduces costs.

Response: The text was modified

Whole Introduction.

Since the authors in the title and for in the aim of research emphasize the breed of pigs - the introduction should contain information why this breed was chosen for research. Whole Introduction. It should be more emphasized what is new in the article (in relation to other research on the inclusion of olive cakes in pig nutrition).

Response: The text was modified

Line 86. N = 8 for one food group is not enough. With such limited research material, we can rather talk about preliminary research.

Response: I apologize but We do not agree. 8 animals/treatment in a field trial for a total of 40 animals feed controlled and 40 carcasses analyzed 40 muscle samples chemical as well analyzed in triplicated is not “limited research material”! We consider that the study has the scientific data necessary to reach statistical valid conclusions.

Chapter 2.1. It should be clarified whether the animals were of the same age at the beginning of the experiment, whether the animals were fed the same until the beginning of the experiment (i.e. until the mass was 100 kg), if so, how they were fed, whether the animals were kept in the same conditions , in which? If the animals were not initially aligned then the experiment should be considered as incorrectly designed.

Response: Done. The text was clarified.

Lines 230-231; Lines 238-243; Lines 251-266; Lines 277-284; Lines 289-301; Lines 331-336; many parts of a paragraph about the chemical composition of meat; and Lines 373-380; Lines 407-412 - many fragments with the composition of fatty acids: The discussion has nothing to do with the purpose of the research. It is unnecessary in this form. The discussion should be focused on the purpose of the research.

Response: The discussion was built according to the data obtained and the main propose of the study.

Minor comments:

Line 2. The title makes reference to the quality of the meat - and there is no reference to the quality of the fat - the title should be reworded

Response: Suggestion accepted; changes have been made in the revised version manuscript.

Lines 26-27. The types of feed should be listed in the order in which they appear in the tables (e.g. table 1)

Response: Suggestion accepted; changes have been made in the revised version of the manuscript.

Lines 32-33. This sentence is incomprehensible.

Response: Suggestion accepted; The phrase has been removed so as not to create entropy

Line38. In order to provide "as a general conclusion" information on the quality of fat in the abstract, information on this subject should be provided earlier in the text.

Response: The text was modified according to the referee suggestion.

Lines 41-42. Keywords it is worth expanding with the terms concerning the other tested material properties. Line 86. The breed of pigs must be indicated

Response: Suggestion accepted; changes have been made in the revised version of the manuscript.

Line 87. Explain why this particular amount of olive oil cakes was added to the feed.

Response: Suggestion accepted; changes have been made in the revised version of the manuscript

Lines 87-88. It should be clarified whether the feeds were balanced in terms of the content of basic ingredients (protein, fat, carbohydrates)? What was the content of the basic chemicals in the individual feeds? Table 1. Why the weight of the feed ingredients does not sum up to 100,00

Response: Suggestion accepted; changes have been made in the revised version of the manuscript (table 1)

Lines 89-90. The highest total PUFA value was found in the control feed, why? why do the authors omit it and how can this be explained? Moreover, the differences in the SFA are very small. Generally, due to the small differences in the quoted values - without a statistical analysis, it cannot be said that the values were higher.

Response: Done. The text was modified

Line 114 At what minute after slaughter, the first pH measurement was made (this is important due to the quality of the meat). What muscle was the pH measured in?

Response: 1 hour and 24 h hours after slaughter as it was pointed out in Table 3; pH was assessed over the Longissimus Thoraces et lumborum muscle, between the 7th and the 10th vertebrae

Line 118. How long after slaughter was the color measured? What conditions were applied (bloming time?)

Response: Color measurement is taken fifteen minutes after cutting the piece

Line 119, Line 286. redness (a*), and yellowness (b*) – this is a simplification - redness and yellowness only apply to positive values of these parameters.

Response: Ok. Corrected

Line 137. Are the measurements of the carcass length, limbs etc. are essential in the experiment?

Response: Yes once it would be important to give some information useful for a putative commercial carcass grading system for this breed.

Is there still a clear development of the skeletal system at this stage of the pig's life, when the experiment was carried out?

Response: The body weight of animals studied was 134 - 143 kg and the Body weight of adult males is around 180 kg and 150 kg for females…

Line 155 – please add „and fat”

Response: Suggestion accepted; changes have been made in the revised version of the manuscript.

Line 163 – What was the size of the heated sample. Was the temperature of the medium the same as the end temperature of the sample? (Usually, the temperature of the heating medium is a few degrees higher).

Response: Suggestion accepted; changes have been made in the revised version of the manuscript (line 163). It is mentioned that the thermal center of the sample remains in the bath until reaching 70 ºC

Lines 165 -166. What was the time from warming up to measuring the cutting force? What was the temperature of the samples at the time of testing?

Response: In lines 165-166, it refers to the average time for the realization of the cutting force

Line 192; for *P < 0.05, **P < 0.01 or ***P < 0.001 significance levels – it is unnecessary - the authors only have in the tables / figures *P < 0.05.

Response: Suggestion accepted; changes have been made in the revised version of the manuscript.

Lines 193-194: For the sensorial statistical analysis of the results, an analysis of variance (ANOVA) of one-way using IBM SPSS Statistics 22.0 program (IBM Corporation, Somers, NY, USA) was performed for variables considered in the study…. There is no sensory research in the article.

Response: Suggestion accepted; changes have been made in the revised version of the manuscript.

Line 221. Figure 2 should be supplemented with information on the statistical analysis (symbols in the figures, information in the title of the figure). Figures 3, 4,. Tables 2, 3, 4, 5, 6, In the captions, information about the statistical analysis should be provided (what the letter symbols mean), In the captions under the tables, it should be written what SE means. Figure 2, 3, 4. Axes should start at 0. Figure 4. The drawing is illegible. All tables and figures: To make it easier to read the article, it is worth giving an explanation of abbreviations T1, T2, T3, T4, T5 under the individual tables and figures (now they are only in the methodology under table 1). Figures 3, 4. Why is there no standard deviation in these figures (in Figure 2 there is).

Response: Suggestions accepted; changes have been made in the revised version of the manuscript. Regarding the axes, we consider it appropriate that they do not start at zero to make the figure more visible.

Lines 223 -224 Same sentence twice? Moreover, what does it mean depth at the level of the 7th rib? No detailed explanation in the methodology - should be completed.

Response: Suggestion accepted; changes have been made in the revised version of the manuscript.

Lines 225-226. What does “length of the muscle” mean? No detailed explanation in the methodology - should be completed. Instead of length of the muscle, it is better to measure and discuss the cross-sectional area of the muscle.

Response: The text was clarified

Lines 228, 229 The values vary between ... .. it should be added that "mean values"

Response: Suggestion accepted; changes have been made in the revised version of the manuscript.

Line 249 Information on backfat thickness has not been provided in detail in the methodology. In addition, what was the purpose of checking backfat thickness at so many points (in 9) - did the authors expect that a feed modification would change backfat thickness at only one point in the carcass? Table 3. The table methodology does not specify what Dressing percentage (%) means? Tables 2-5 The form of the titles should be standardized. Now there are different forms, e.g. Effect of different olive cake treatments on …. or e.g.. Chemical composition … for the 5 diet treatments

Response: Done. The text was modified.

Did the authors expect that a feed modification would change backfat thickness at only one point in the carcass? We don't understand what the referee wants to mean

Line 348. The title is incorrect WHC and SF it is not a chemical composition.

Response: Suggestion accepted; changes have been made in the revised version of the manuscript.

Lines 323-324: On what basis do the authors claim that “The lower content of oleic acid in the control diet (T1) in comparison with the other treatments (see Table 1) could be the origin of these differences.” Do other authors also point to the influence of this fatty acid?

Response: deleted phrase. Writing error.

Lines 352-353: The sentence “It was interesting to note that the effect of the diet treatment on moisture had an inverse trend to intramuscular fat content.” does not indicate an "interesting" relationship, but an obvious relationship between the chemical components in the meat.

Response: Suggestion accepted; changes have been made in the revised version of the manuscript.

Line 372. The T3 had a significantly higher WHC value than T4 – abbreviation T3 and T4 should be replaced with text.

Response: As a matter of homogeneity, we chose to use acronyms throughout the text (from T1 to T5). Making this difference does not seem to us the most appropriate.

Lines 34-35; 380-381: On what basis do the authors claim that: The values of water holding capacity and shear force observed conform to a non-exudative and firmness meat and The values of WHC and SF observed conform to a non-exudative and firm meat.?

Response: The WHC and SF associated to pH and CIELAB color, as we explain before, in comparison with the values of the cited works conform to a non-exudative and firm meat.

Lines 388-391; 458-461. In my opinion, as the differences are not statistically significant, it is not worth mentioning. The differences the authors write about are rather small - so there is no sense in pointing to the existence of a tendency.

Response: OK

Line 478. In case of margaric acid (C17:1n-7) – all samples are marked with „

Response: We don't understand what the referee wants to mean with „

Lines 523-524. On what basis the authors write such a statement?

Response: Suggestion accepted; changes have been made in the revised version of the manuscript.

Lines 534-548. In my opinion, these sentences are unnecessary ???

Response: OK, however we think it will be interesting give some ways of future work needed.

Reviewer 3 Report

Manuscript Number: foods-1733234, titled:

Can the introduction of different olive cakes affect the carcass and meat quality of Bísaro pork?

Review 1 – 7 May 2022

Dear Editor of Foods

the argument is interesting but needs to be improved. The introduction section has to be widely extended and well argued in relation of the aim of the work. The Materials and Methods section has to be detailed and well explained. Inaccuracies in the manuscript. The statements of the authors have to be supported by proper references. The references section has to be re-arranged as suggested in the instructions for Authors of Foods.

I suggest a major revision

To the Authors (in detail):

  1. the argument is interesting but needs to be improved. The introduction section has to be widely extended and well argued in relation of the aim of the work. The Materials and Methods (M&M) section has to be detailed and well explained. Inaccuracies in the manuscript. The statements of the authors have to be supported by proper references. The references section has to be re-arranged as suggested in the instructions for Authors of Foods;

2. Introduction section, lines 48-56, insert a percentage composition of each component (including oil) also related to the two or three phases extraction systems. Insert also some proper references. Studies related with this argument;

3. Introduction section explain that the olive composition is related to many factors such as: cultivar, harvest date, harvest year, geographical area of production, irrigation, fertilization and so on. Support this statement with some proper reference about pre-and post-harvest factors influencing olive and olive oil;

4. Introduction section, discuss better in relation with the olive composition: fiber, sugars, phenols, oil and support this part of the introduction with proper references;

5. Introduction section, your study is focused on olive and olive oil but nothing you have written about the olive oil composition. Please, explain that OO is composed by: 98-98.5% triglycerides and 1.5-2% minor components such as: sterols; fatty alcohols ; waxes ; phenols ; tocopherols, carotenoids, squalene [1-4]. Specify that the olive oil composition is related with many parameters (cv, harvest date, harvest year, geographical area of production). Fatty acids are a part of triglycerides. Support this statement with proper references. Include each reference after each class of compounds and do not cumulate the references at the end of the sentence. Please, find, read and discuss:

[1] Influence of cultivar and harvest year on triglyceride composition of olive oils produced in Calabria (Southern Italy).

European Journal of Lipid Science and Technology, 115 (8) 928-934 (2013)

[2] Sterols and Triterpene Diols in Virgin Olive Oil: A Comprehensive Review on Their Properties and Significance, with a Special Emphasis on the Influence of Variety and Ripening Degree.

Horticulturae 2021, 7, 493. https://doi.org/10.3390/horticulturae7110493

[3] Geographical Characterization of Olive Oils from the North Aegean Region Based on the Analysis of Biophenols with UHPLC-QTOF-MS.

Foods 2021, 10, 2102. https://doi.org/10.3390/foods10092102

[4] Simultaneous Determination of Pigments, Tocopherols, and Squalene in Greek Olive Oils: A Study of the Influence of Cultivation and Oil-Production Parameters.

Foods 2020, 9, 31; doi:10.3390/foods9010031

6. Introduction section, Lines 72-74, specify that the SFA reduction and the increase in MUFA is mainly due to the triglyceride composition of olive oil. Support this statement with some reference;

7. 2.1 sub-section: indicate the breed of pigs; how many males and how many females;

8. 2.1 sub-section, no information is given about the olive cultivar/s of the cake used in your experiment.

9. M&M section. No information is given about the biometrics of these cultivar, because the cake composition is also related to fruits weight, pulp weight, pulp/pit ratio, water content, oil content, and so on. The olive cake you have used is different with respect the olive cake used in another experiment and this has influenced the results of your experiment. Before this, in the introduction section, discuss about biometrics of drupes and support the discussion in the introduction section about the biometric evaluation of olive cultivars and support your discussion with some proper reference;

10. M&M section. You have to include one section to well detail the thesis you have used in your experiment. Describe the feed composition and olive oil content in each thesis and the type of oil used in your T5 (see lines 458-459). Table 1 is not enough;

11. M&M section no information is given about the analysis of the experimental diet. This part has to be detailed with analytical methods;

12. 2.4 sub-section and in the whole manuscript, tables figures, when you indicate a temperature, separate the numeric value from the symbol; 70 °C and not 70°C;

13. 2.4 sub-section, line 180, 2.35 mL. It is not necessary to include zero as third decimal;

14. 2.4 sub-section, line 184, when you indicate the column length, write 100 m length. Write length for consistency with other parameters;

15. The one-way ANOVA results are missed in the figure 2;

16. The one way parameters are missed in the caption of figure 3;

17. 3.1 and 2.5 sub-sections: please, when you indicate the significance, use always the same criterion in relation of spacing between letter, symbol and numeric value: P < 0.05 or P<0.05? Please, be consistent throughout the manuscript, figures and tables;

18. 3.1 sub-section, line 234, and in the whole manuscript: please separate the numeric value from the symbol: 9.92 cm and not 9.92cm;

19. Figure 4, the bars, letters and numbers cannot be well read because are too faint. Please, re-arrange the color also in all figures. In addition, letters and numbers have a too small size. Try to print and to read;

20.3.1 sub-section, line 247: Verify the English language and re-write this sentence;

21. Caption of table 2, the significance is missed n.s. for P≥ 0.05;

22. Table 3 and throughout the manuscript: L*, a*, b*; have to be italicized;

23. Caption of table 3, the significance is missed n.s. for P≥ 0.05;

24. 3.2 sub-section and in the whole manuscript, sometime you have included one space between the numeric value and the symbol (5.22 %) and sometime non (7.22%), please, be consistent in the whole manuscript;

25. Caption of table 4 (p in small letter for the significance) but in other part of your manuscript you have used the capital letter. Please, be consistent in the whole manuscript, tables, figures;

26. 3.3 sub-section, line 387: palmitic and stearic and not stearic and palmitic: either because 16 is higher than 18 or because 16 is detected before 18;

27. 3.3 sub-section, line 420: Mangalitsa in capital letter;

28. Table 5, the sum of MUFA, PUFA and SFA is not 100 in the column T3 and T5. Please, verify data;

29. 3.3 section, line 405 (g/kg), whereas 3.2 sub-section, line 328 (g kg-1): fraction or exponent? Please, be consistent in the whole manuscript. In addition, separate the numeric value from the unit: 100 g and not 100g (line 405);

30. Table 6, the sum of MUFA, PUFA and SFA is not 100 in the column T3 and T5. Please, verify data

31. References section. This section is not arranged as per Foods instructions for Authors. For example; the journal name has to be italicized and abbreviated (ref 1); the volume number is missed (ref 1); The journal name in capital letters (ref 1);

32. Please, write in blue color or evidence differently the corrections you will do

I suggest a major revision

Regards.

Author Response

Dear review,

All modifications were made following the reviewer's suggestions and comments, and responses to their comments are also attached. Thanks to their recommendations, significant modifications were made throughout the manuscript.

Thank you for your attention.

Answers to Reviewer 2

  1. the argument is interesting but needs to be improved. The introduction section has to be widely extended and well argued in relation of the aim of the work. The Materials and Methods (M&M) section has to be detailed and well explained. Inaccuracies in the manuscript. The statements of the authors have to be supported by proper references. The references section has to be re-arranged as suggested in the instructions for Authors of Foods;
  2. Introduction section, lines 48-56, insert a percentage composition of each component (including oil) also related to the two or three phases extraction systems. Insert also some proper references. Studies related with this argument;

Response 1 and 2: Suggestions accepted; changes have been made in the revised version of the manuscript.

  1. Introduction section explain that the olive composition is related to many factors such as: cultivar, harvest date, harvest year, geographical area of production, irrigation, fertilization and so on. Support this statement with some proper reference about pre-and post-harvest factors influencing olive and olive oil;
  2. Introduction section, discuss better in relation with the olive composition: fiber, sugars, phenols, oil and support this part of the introduction with proper references;
  3. Introduction section, your study is focused on olive and olive oil but nothing you have written about the olive oil composition. Please, explain that OO is composed by: 98- 98.5% triglycerides and 1.5-2% minor components such as: sterols; fatty alcohols ; waxes ; phenols ; tocopherols, carotenoids, squalene [1-4]. Specify that the olive oil composition is related with many parameters (cv, harvest date, harvest year, geographical area of production). Fatty acids are a part of triglycerides. Support this statement with proper references. Include each reference after each class of compounds and do not cumulate the references at the end of the sentence. Please, find, read and discuss:

[1] Influence of cultivar and harvest year on triglyceride composition of olive oils produced in Calabria (Southern Italy). European Journal of Lipid Science and Technology,

[2] Sterols and Triterpene Diols in Virgin Olive Oil: A Comprehensive Review on Their Properties and Significance, with a Special Emphasis on the Influence of Variety and Ripening Degree. Horticulturae 2021, 7, 493. https://doi.org/10.3390/horticulturae7110493

[3] Geographical Characterization of Olive Oils from the North Aegean Region Based on the Analysis of Biophenols with UHPLC-QTOF-MS. Foods 2021, 10, 2102. https://doi.org/10.3390/foods10092102

[4] Simultaneous Determination of Pigments, Tocopherols, and Squalene in Greek Olive Oils: A Study of the Influence of Cultivation and Oil-Production Parameters. Foods 2020, 9, 31; doi:10.3390/foods9010031

Response 3, 4 and 5: The objective of the present study was not to study the characteristics of olive cultivars and olive oil as mentioned in items 3, 4 and 5. Nor was it to study different extraction methods. In fact, the objective was to study the effect on carcass and meat quality in pigs fed a by-product of olive oil extraction. It is out of the question to study the influences of cultivars, harvest years or olive oil compositions. We appreciate the references indicated by the referee, but we think they are out of the context of the present study.

  1. Introduction section, Lines 72-74, specify that the SFA reduction and the increase in MUFA is mainly due to the triglyceride composition of olive oil. Support this statement with some reference;

Response 6: Suggestion accepted, changing the place of references.

  1. 2.1 sub-section: indicate the breed of pigs; how many males and how many females;

Response 7: Suggestion accepted; changes have been made in the revised version of the manuscript.

  1. 2.1 sub-section, no information is given about the olive cultivar/s of the cake used in your experiment.
  2. M&M section. No information is given about the biometrics of these cultivar, because the cake composition is also related to fruits weight, pulp weight, pulp/pit ratio, water content, oil content, and so on. The olive cake you have used is different with respect the olive cake used in another experiment and this has influenced the results of your experiment. Before this, in the introduction section, discuss about biometrics of drupes and support the discussion in the introduction section about the biometric evaluation of olive cultivars and support your discussion with some proper reference;
  3. M&M section. You have to include one section to well detail the thesis you have used in your experiment. Describe the feed composition and olive oil content in each thess and the type of oil used in your T5 (see lines 458- 459). Table 1 is not enough.

Response 8, 9 and 10: More information was added to the text. However, the present work are part of the project research “BisOlive: Use of olive pomace in the feeding of Bísaro swine. Evaluation of the effect on meat quality” project. NORTE-01-0247-FEDER-072234 and the olive cakes were purchased from different olive oil industries that receive olives from different farms and as we have already said the project did not had an objective to study olive cultivars and we do not have the data you are asking for.

  1. M&M section no information is given about the analysis of the experimental diet. This part has to be detailed with analytical methods.

Response 11: Suggestion accepted; changes have been made in the revised version of the manuscript

  1. 2.4 sub-section and in the whole manuscript, tables figures, when you indicate a temperature, separate the numeric value from the symbol; 70 °C and not 70°C;

Response 12: Suggestion accepted; changes have been made in the revised version of the manuscript.

  1. 2.4 sub-section, line 180, 2.35 mL. It is not necessary to include zero as third decimal;

Response 13: Suggestion accepted; changes have been made in the revised version of the manuscript.

  1. 2.4 sub-section, line 184, when you indicate the column length, write 100 m length. Write length for consistency with other parameters;

Response 14: Suggestion accepted; changes have been made in the revised version of the manuscript.

  1. The one-way ANOVA results are missed in the figure 2;

Response 15: Suggestion accepted; changes have been made in the revised version of the manuscript.

  1. The one way parameters are missed in the caption of figure 3

Response 16: Suggestion accepted; changes have been made in the revised version of the manuscript.

  1. 3.1 and 2.5 sub-sections: please, when you indicate the significance, use always the same criterion in relation of spacing between letter, symbol and numeric value: P < 0.05 or P<0.05? Please, be consistent throughout the manuscript, figures and tables;

Response 17: Suggestion accepted; changes have been made in the revised version of the manuscript.

  1. 3.1 sub-section, line 234, and in the whole manuscript: please separate the numeric value from the symbol: 9.92 cm and not 9.92cm;

Response 18: Suggestion accepted; changes have been made in the revised version of the manuscript.

  1. Figure 4, the bars, letters and numbers cannot be well read because are too faint. Please, re-arrange the color also in all figures. In addition, letters and numbers have a too small size. Try to print and to read;

Response 19: Suggestion accepted; changes have been made in the revised version of the manuscript.

20.3.1 sub-section, line 247: Verify the English language and re-write this sentence;

Response 20: Suggestion accepted; changes have been made in the revised version of the manuscript. (line 251-252)

  1. Caption of table 2, the significance is missed n.s. for P≥ 0.05;

Response 21: Suggestion accepted; changes have been made in the revised version of the manuscript.

  1. Table 3 and throughout the manuscript: L*, a*, b*; have to be italicized;

Response 22: Suggestion accepted; changes have been made in the revised version of the manuscript.

  1. Caption of table 3, the significance is missed n.s. for P≥ 0.05;

Response 23: Suggestion accepted; changes have been made in the revised version of the manuscript.

  1. 3.2 sub-section and in the whole manuscript, sometime you have included one space between the numeric value and the symbol (5.22 %) and sometime non (7.22%), please, be consistent in the whole manuscript

Response 24: Suggestion accepted; changes have been made in the revised version of the manuscript.

  1. Caption of table 4 (p in small letter for the significance) but in other part of your manuscript you have used the capital letter. Please, be consistent in the whole manuscript, tables, figures;

Response 25: Suggestion accepted; changes have been made in the revised version of the manuscript.

  1. 3.3 sub-section, line 387: palmitic and stearic and not stearic and palmitic: either because 16 is higher than 18 or because 16 is detected before 18;

Response 26: Suggestion accepted; changes have been made in the revised version of the manuscript.

  1. 3.3 sub-section, line 420: Mangalitsa in capital letter;

Response 27: Suggestion accepted; changes have been made in the revised version of the manuscript.

  1. Table 5, the sum of MUFA, PUFA and SFA is not 100 in the column T3 and T5. Please, verify data;

Response 28: Suggestion accepted; changes have been made in the revised version of the manuscript.

  1. 3.3 section, line 405 (g/kg), whereas 3.2 sub-section, line 328 (g kg ): fraction or exponent? Please, be consistent in the whole manuscript. In addition, separate the numeric value from the unit: 100 g and not 100g (line 405);

Response 29: Suggestion accepted; changes have been made in the revised version of the manuscript.

  1. Table 6, the sum of MUFA, PUFA and SFA is not 100 in the column T3 and T5. Please, verify data

Response 30: Suggestion accepted; changes have been made in the revised version of the manuscript.

  1. References section. This section is not arranged as per Foods instructions for Authors. For example; the journal name has to be italicized and abbreviated (ref 1); the volume number is missed (ref 1); The journal name in capital letters (ref 1);

Response 31: Suggestion accepted; changes have been made in the revised version of the manuscript.

  1. Please, write in blue color or evidence differently the corrections you will do I suggest a major revision

Response 32: The changes have been recorded in the document

Round 2

Reviewer 2 Report

Manuscript Number: foods-1733234-peer-review-v1

Title: Can the introduction of different olive cakes affect the carcass, meat and fat quality of Bísaro pork?

Authors: Ana Leite, Rubén Domínguez, Lia Vasconcelos, Iasmin Ferreira, Etelvina Pereira, Victor Pinheiro, D. Outor-Monteiro, Sandra Rodrigues, José Manuel Lorenzo, Eva María Santos, Silvina Cecilia Andrés, Paulo C. B. Campagnol, and Alfredo Teixeira

Overview and general recommendation:

The manuscript has been revised in many areas. Unfortunately, many of the comments I made in the first review were still not taken into account by the Authors (see detailed comments below). 

Major comments:

In abstract (lines 31-33) and in the text (lines 619-623) the authors indicate that based on the results of pH and color parameters measurements, it can be concluded that the tested meat was free from PSE and DFD defects, without providing any references. Information on how, on the basis of the pH results and color parameters, the classification of the raw material for good quality meat and with PSE and DFD defects was carried out should be included in the methodology (when describing the meat quality assessment methods). Specific classification values should be provided.

Lines 34-35; 1110-1111: On what basis do the authors claim that: “The values of water holding capacity and shear force observed conform a non-exudative and firm meat” and “The values of WHC and SF observed conform to a non-exudative and firm meat”. What specific values indicate this? From what level of WHC and SF values is meat already considered "exudative" or "firm"? Please provide source or delete / edit text.

Figures 1, 2, 3 are completely illegible. Due to the illegibility of figures 2 and 3, it is not possible to check the correctness of the text with the values presented in the figure.

The manuscript should include a chapter on the impact of dietary modification on the characteristics of meat (this chapter should be included after the description of carcass quality characteristics). This chapter should include information about the pH, colour parameters, WHC, SF (these are the characteristics of meat quality). Now the pH and colour parameters are in the chapter on the characteristics of carcasses and WHC and SF in the chapter on chemical composition (after all, WHC and SF are not chemical composition!). A separate table should be created for these results (pH, colour, WHC, SF). The chemical composition of meat can be given by the meat quality characteristics (chemical composition also belongs to the meat quality characteristics) or given as a separate subchapter.

Lines 416-457; 471-473; 485-487, 613-615,  623-635, 863-868, 884-895, 1104-1109, 1359-1366, 1375…,1563-1565, The discussion about the influence of breed on the examined features should be minimized and the text should be supplemented or edited so that the discussion refers to the influence of feeding on the features of carcasses and meat (the aim of the research is the influence of modification of pig feeding on the quality of carcasses, meat and fat, not carcass characteristics, meat, fat from Bísaro pigs).

Minor comments:

Title: – “Can the introduction of different olive cakes affect the carcass, meat and fat quality of Bísaro pork? Pork – it is meat (not carcass / fat). Title should be redrafted: Can the introduction of different olive cakes affect the Bísaro carcass, meat and fat quality?

Line 25-28.  The sentence „The present work aims to evaluate the inclusion of different olive cakes in the feed of Bisaro pigs” should be redrafted (lack - what was the effect of the olive cakes feed additive tested)

Line 38. In order to provide „as a general conclusion” information on the quality of fat in the abstract, information on this subject should be provided earlier in the text. There is nothing in the abstract about lipidic quality.

Figures 2, 3, 4. The axes should start with the value 0. The presented graphs emphasize visually differences that do not exist.

Figure 3. There is no information about the statistical analysis in the description under the figure.

Line 140. Forty pigs weighing around 100 kg body weight ….. the name of the breed should be added: Forty Bísaro pigs weighing around 100 kg body weight…

Line 143. The animals were separated by batches of 10…. If there were 40 pigs and they were divided into 5 groups - it should be 8. The description is incomprehensible.

Lines 192-194. “there were no statistically significant differences for MUFA, PUFA and SFA content of the diets incorporating olive cakes in relation to the control diet (T1)”. There is no information in the text about the statistical analysis and the number of repetitions of the research. Did the lack of differences apply to all fatty acids? For example, the differences in PUFA between T1 and T2 were quite large - were they not statistically significant? The text should be supplemented with this information. What is the reason for such a high proportion of PUFA in the T1 feed?

Line 224. Pigs were slaughtered at an average body weight of 135 ± SD kg – There is no SD value. Must be completed.

Line 338. The text should include information about the time after which the colour was measured after slaughter? What conditions were used (blooming time?)

Lines 340 – 342 . Why there is h and not H* in the equation?

Line 333. The text should be supplemented with information after what time after slaughter the first pH measurement was made (methodological information should be provided in the methodology and not in the table - the information in the table is only supplementary).

Table 2 contains information about Carcass depth and Leg fat thickness – the methodology does not contain information about these measurements. The text should be completed.

Line 357. The text should include information about the temperature of the samples at the time of testing shear force

Lines 466-467. No significant differences were observed in the depth at the level of the 7th rib and the mean values obtained vary between 3.59 and 4.09 cm. (the sentence does not specify what “depth”). The text should be redrafted.

Line 520. No significantly differences were observed (p>0.05). … there is no information between what were “no significant differences observed”.  The text should be redrafted.

Lines 1756-1757. Significant differences (p<0.05) between treatments were found in palmitoleic acid (C16:1n-7), margaric acid (C17:1n-7), and eicosenoic acid (C20:1n-9). No – in case of margaric acid (C17:1n-7) there are no statistically significant differences (all means are marked "a").

Lines 1118-1120; 1581-1585. In my opinion, as the differences are not statistically significant, it is not worth mentioning. The differences which the authors write about are rather small - so there is no basis for pointing the existence of any tendency.

Lines 1981-1985. In my opinion, these sentences are unnecessary.

Author Response

Dear review,

All modifications were made following the reviewer's suggestions and comments, and responses to their comments are also attached. Thanks to their recommendations, significant modifications were made throughout the manuscript.

Thank you for your attention.

Answers to Reviewer 2

Overview and general recommendation:

The manuscript has been revised in many areas. Unfortunately, many of the comments I made in the first review were still not taken into account by the Authors (see detailed comments below).

Major comments:

In abstract (lines 31-33) and in the text (lines 619-623) the authors indicate that based on the results of pH and color parameters measurements, it can be concluded that the tested meat was free from PSE and DFD defects, without providing any references. Information on how, on the basis of the pH results and color parameters, the classification of the raw material for good quality meat and with PSE and DFD defects was carried out should be included in the methodology (when describing the meat quality assessment methods). Specific classification values should be provided.

Response: Suggestions accepted; changes have been made in the revised version of the manuscript (in yellow color and the removed text)

Lines 34-35; 1110-1111: On what basis do the authors claim that: “The values of water holding capacity and shear force observed conform a non-exudative and firm meat” and “The values of WHC and SF observed conform to a non-exudative and firm meat”. What specific values indicate this? From what level of WHC and SF values is meat already considered "exudative" or "firm"? Please provide source or delete / edit text.

Response: Suggestions accepted; changes have been made in the revised version of the manuscript (in yellow color and the removed text)

Figures 1, 2, 3 are completely illegible. Due to the illegibility of figures 2 and 3, it is not possible to check the correctness of the text with the values presented in the figure.

Response: Suggestions accepted; changes have been made in the revised version of the manuscript

The manuscript should include a chapter on the impact of dietary modification on the characteristics of meat (this chapter should be included after the description of carcass quality characteristics). This chapter should include information about the pH, colour parameters, WHC, SF (these are the characteristics of meat quality). Now the pH and colour parameters are in the chapter on the characteristics of carcasses and WHC and SF in the chapter on chemical composition (after all, WHC and SF are not chemical composition!). A separate  table should be created for these results (pH, colour, WHC, SF). The chemical composition of meat can be given by the meat quality characteristics (chemical composition also belongs to the meat quality characteristics) or given as a separate subchapter.

Response: Suggestions accepted; changes have been made in the revised version of the manuscript (in yellow color and the removed text)

Lines 416-457; 471-473; 485-487, 613-615, 623-635, 863-868, 884-895, 1104-1109, 1359-1366, 1375…,1563-1565, The discussion about the influence of breed on the examined features should be minimized and the text should be supplemented or edited so that the discussion refers to the influence of feeding on the features of carcasses and meat (the aim of the research is the influence of modification of pig feeding on the quality of carcasses, meat and fat, not carcass characteristics, meat, fat from Bísaro pigs).

Response: Suggestions accepted; changes have been made in the revised version of the manuscript (revised text)

Minor comments:

Title: – “Can the introduction of different olive cakes affect the carcass, meat and fat quality of Bísaro pork? Pork – it is meat (not carcass / fat). Title should be redrafted: Can the introduction of different olive cakes affect the Bísaro carcass, meat and fat quality?

Response: Suggestions accepted; changes have been made in the revised version of the manuscript (in yellow color in the manuscript)

Line 25-28. The sentence „The present work aims to evaluate the inclusion of different olive cakes in the feed of Bisaro pigs” should be redrafted (lack - what was the effect of the olive cakes feed additive tested)

Response: Suggestions accepted; changes have been made in the revised version of the manuscript (in yellow color in the manuscript)

Line 38. In order to provide „as a general conclusion” information on the quality of fat in the abstract, information on this subject should be provided earlier in the text. There is nothing in the abstract about lipidic quality.

Response: Suggestions accepted; changes have been made in the revised version of the manuscript (in yellow color in the manuscript)

Figures 2, 3, 4. The axes should start with the value 0. The presented graphs emphasize visually differences that do not exist.

Response: Suggestions accepted; changes have been made in the revised version of the manuscript (the figures)

Figure 3. There is no information about the statistical analysis in the description under the figure.

Response: Suggestions accepted; changes have been made in the revised version of the manuscript (in yellow color in the manuscript)

Line 140. Forty pigs weighing around 100 kg body weight ….. the name of the breed should be added: Forty Bísaro pigs weighing around 100 kg body weight…

Response: Suggestions accepted; changes have been made in the revised version of the manuscript (in yellow color in the manuscript)

Line 143. The animals were separated by batches of 10…. If there were 40 pigs and they were divided into 5 groups - it should be 8. The description is incomprehensible.

Response: Suggestions accepted; changes have been made in the revised version of the manuscript (in yellow color in the manuscript)

Lines 192-194. “there were no statistically significant differences for MUFA, PUFA and SFA content of the diets incorporating olive cakes in relation to the control diet (T1)”. There is no information in the text about the statistical analysis and the number of  repetitions of the research. Did the lack of differences apply to all fatty acids? For example, the differences in PUFA between T1 and T2 were quite large - were they not statistically significant? The text should be supplemented with this information. What is the reason for such a high proportion of PUFA in the T1 feed?

Response: The experimental feed trial was carried on Trás-os-Montes e Alto Douro University, Vila Real – Portugal. This was the technical specification information of the reason that we were given. (in yellow color in the manuscript)

Line 224. Pigs were slaughtered at an average body weight of 135 ± SD kg – There is no SD value. Must be completed

Response: Suggestions accepted; changes have been made in the revised version of the manuscript (in yellow color)

Line 338. The text should include information about the time after which the colour was measured after slaughter? What conditions were used (blooming time?)

Response: Suggestions accepted; changes have been made in the revised version of the manuscript (in yellow color)

Lines 340 – 342 . Why there is h and not H* in the equation?

Response: Suggestions accepted; changes have been made in the revised version of the manuscript (in yellow color)

Line 333. The text should be supplemented with information after what time after slaughter the first pH measurement was made (methodological information should be provided in the methodology and not in the table - the information in the table is only supplementary).

Response: Suggestions accepted; changes have been made in the revised version of the manuscript (in yellow color)

Table 2 contains information about Carcass depth and Leg fat thickness – the methodology does not contain information about these measurements. The text should be completed.

Response: Suggestions accepted; changes have been made in the revised version of the manuscript (in yellow color and figure 1)

Line 357. The text should include information about the temperature of the samples at the time of testing shear force

Response: Suggestions accepted; changes have been made in the revised version of the manuscript (in yellow color)

Lines 466-467. No significant differences were observed in the depth at the level of the 7th rib and the mean values obtained vary between 3.59 and 4.09 cm. (the sentence does not specify what “depth”). The text should be redrafted.

Response: Suggestions accepted; changes have been made in the revised version of the manuscript (figure 2)

Line 520. No significantly differences were observed (p>0.05). … there is no information between what were “no significant differences observed”. The text should be redrafted.

Response: Suggestions accepted; changes have been made in the revised version of the manuscript

Lines 1756-1757. Significant differences (p<0.05) between treatmenst were found in palmitoleic acid, margaric acid and eicosenoic acid. No – in case of margaric acid there are no statiscally significant differences (all means are marked “a”)

Response: Suggestions accepted; changes have been made in the revised version of the manuscript

Lines 1118-1120; 1581-1585. In my opinion, as the differences are not statistically significant, it is not worth mentioning. The differences which the authors write about are rather small - so there is no basis for pointing the existence of any tendency.

Response: Suggestions accepted; texts have been removed.

Lines 1981-1985. In my opinion, these sentences are unnecessary.

Response: Suggestions accepted; the conclusion section has been revised

Reviewer 3 Report

Manuscript Number: foods-1733234, titled:

Can the introduction of different olive cakes affect the carcass and meat quality of Bísaro pork?

Review 2 – 19 May 2022

Dear Editor of Foods

the argument is interesting but needs to be improved. The introduction section has to be widely extended and well argued in relation of the aim of the work. The statements of the authors have to be supported by proper references. The references section has to be re-arranged as suggested in the instructions for Authors of Foods.

I suggest a major revision

To the Authors (in detail):

  1. Introduction section, insert a percentage composition of each component (including oil) also related to the two or three phases extraction systems. Insert also some proper references of studies related with this argument. Discuss also the relation between the extraction system and fruits. Support this with proper references;

  1. Introduction section explain that the olive fruit composition is related to many factors such as: cultivar, harvest date, harvest year, geographical area of production, irrigation, fertilization and so on. Support this statement with some proper references about pre-and post-harvest factors influencing olive and olive oil;

  1. Introduction section, discuss better in relation with the olive composition: fiber, sugars, phenols, oil and support this part of the introduction with proper references;

  1. Introduction section, your study is focused on olive and olive oil but nothing you have written about the olive oil composition. Please, explain that OO is composed by: 98-98.5% triglycerides and 1.5-2% minor components such as: sterols; fatty alcohols; waxes; phenols; tocopherols, carotenoids, squalene; n-alkanes and n-alkenes. Specify that the olive oil composition is related with many parameters (cv, harvest date, harvest year, geographical area of production). Fatty acids are a part of triglycerides. Support this statement with proper references. Include each reference after each class of compounds and do not cumulate the references at the end of the sentence. Please, find, read and discuss:

  1. Introduction section, Lines 72-74, specify that the SFA reduction and the increase in MUFA is mainly due to the triglyceride composition of olive oil. Support this statement with some reference;

  1. 2.1 sub-section, no information is given about the olive cultivar/s of the cake used in your experiment.

  1. M&M section. No information is given about the biometrics of these cultivar, because the cake composition is also related to fruits weight, pulp weight, pulp/pit ratio, water content, oil content, and so on. The olive cake you have used is different with respect the olive cake used in another experiment and this has influenced the results of your experiment. Before this, in the introduction section, discuss about biometrics of drupes and support the discussion in the introduction section about the biometric evaluation of olive cultivars and support your discussion with some proper reference;

  1. M&M section. You have to include one section to well detail the thesis you have used in your experiment. Describe the feed composition and olive oil content in each thesis and the type of oil used in your T5 (see lines 458-459). Table 1 is not enough;

  1. M&M section no information is given about the analysis of the experimental diet. This part has to be detailed with analytical methods;

  1. The figure 2 has to be re-edited because it is not readable. The color is too faint;

  1. The one way parameters are missed in the caption of figure 3;

  1. The figure 3 has to be re-edited because it is not readable. The color is too faint;

  1. References section. This section is not arranged as per Foods instructions for Authors. For example; the journal name has to abbreviated, the year in bold etc , please, read the guidelines and be consistent.

  1. Please, write in blue color or evidence differently the corrections you will do

I suggest a major revision

Regards.

Author Response

Dear review,

All modifications were made following the reviewer's suggestions and comments, and responses to their comments are also attached. Thanks to their recommendations, significant modifications were made throughout the manuscript.

Thank you for your attention.

Answers to Reviewer 3

The argument is interesting but needs to be improved. The introduction section has to be widely extended and well argued in relation of the aim of the work. The statements of the authors have to be supported by proper references. The references section has to be re-arranged as suggested in the instructions for Authors of Foods.

I suggest a major revision

  1. Introduction section, insert a percentage composition of each component (including oil) also related to the two or three phases extraction systems. Insert also some proper references of studies related with this argument. Discuss also the relation between the extraction system and fruits. Support this with proper references;

  1. Introduction section explain that the olive fruit composition is related to many factors such as: cultivar, harvest date, harvest year, geographical area of production, irrigation, fertilization and so on. Support this statement with some proper references about preand post-harvest factors influencing olive and olive oil

  1. Introduction section, discuss better in relation with the olive composition: fiber, sugars, phenols, oil and support this part of the introduction with proper references;

  1. Introduction section, your study is focused on olive and olive oil but nothing you have written about the olive oil composition. Please, explain that OO is composed by: 98-98.5% triglycerides and 1.5-2% minor components such as: sterols; fatty alcohols; waxes; phenols; tocopherols, carotenoids, squalene; n-alkanes and nalkenes. Specify that the olive oil composition is related with many parameters (cv, harvest date,  harvest year, geographical area of production). Fatty acids are a part of triglycerides. Support this statement with proper references. Include each reference after each class of compounds and do not cumulate the references at the end of the sentence. Please, find, read and discuss:

  1. Introduction section, Lines 72-74, specify that the SFA reduction and the increase in MUFA is mainly due to the triglyceride composition of olive oil. Support this statement with some reference;

  1. 1 sub-section, no information is given about the olive cultivar/s of the cake used in your experiment.

  1. M&M section. No information is given about the biometrics of these cultivar, because the cake composition is also related to fruits weight, pulp weight, pulp/pit ratio, water content, oil content, and so on. The olive cake you have used is different with respect the olive cake used in another experiment and this has influenced the results of your experiment. Before this, in the introduction section, discuss about biometrics of drupes and support the discussion in the introduction section about the biometric evaluation of olive cultivars and support your discussion with some proper reference.

  1. M&M section. You have to include one section to well detail the thesis you have used in your experiment. Describe the feed composition and olive oil content in each thesis and the type of oil used in your T5 (see lines 458-459). Table 1 is not enough;

Response 1 to 8:

As previously answered, it is impossible for us to respond adequately to points 1 to 8 placed by the referee. The project that this work is a part of used different types of olive oil cakes from different extraction units that receive olives from all over the Northeast of Portugal. The only analysis performed by us was the lipid profile of each cake. We do not know the origin of the olives of each cake, respective cultivars, harvest dates, whether or not the olive groves were irrigated, fertilized and other forms of management of the cultivars, as well as the composition of the oils from these cake. In fact, our study was based solely on analyzing the physic-chemical quality of meat from animals that were fed with those 4 different types of olive cakes. We understand the referee's interest in these questions, but they are not within the scope of this study, which is not about the quality of olive oil or even the quality of its cake. It is only about the effect of a by-product in the quality of meat.

  1. M&M section no information is given about the analysis of the experimental diet. This part has to be detailed with analytical methods;

Response: The experimental feed trial was carried on Trás-os-Montes e Alto Douro University, Vila Real – Portugal. This was the technical specification information of the reason that we were given. (in yellow color in the manuscript)

  1. The figure 2 has to be re-edited because it is not readable. The color is too faint;

Response: Suggestions accepted; changes have been made in the revised version of the manuscript (in yellow color in the manuscript)

  1. The one way parameters are missed in the caption of figure 3;

Response: Suggestions accepted; changes have been made in the revised version of the manuscript (in yellow color in the manuscript)

  1. The figure 3 has to be re-edited because it is not readable. The color is too faint;

Response: Suggestions accepted; changes have been made in the revised version of the manuscript (in yellow color in the manuscript)

  1. References section. This section is not arranged as per Foods instructions for Authors. For example; the journal name has to abbreviated, the year in bold etc , please, read the guidelines and be consistent

Response: Suggestions accepted; changes have been made in references list of the revised version of manuscript.

  1. Please, write in blue color or evidence differently the corrections you will do

Response: write in yellow color